# POST-HOC REWARD CALIBRATION: A CASE STUDY ON LENGTH BIAS

[1]**Zeyu Huang**  [2]**Zihan Qiu**  [3]**Zili Wang**  [1]**Edoardo M. Ponti**  [1,4]**Ivan Titov**
[1]University of Edinburgh  [2]Alibaba Group  [3]INF Technology  [4]University of Amsterdam
`zeyu.huang@ed.ac.uk`

## ABSTRACT

Reinforcement Learning from Human Feedback aligns the outputs of Large Language Models with human values and preferences. Central to this process is the reward model (RM), which translates human feedback into training signals for optimising LLM behaviour. However, RMs can develop biases by exploiting spurious correlations in their training data, such as favouring outputs based on length or style rather than true quality. These biases can lead to incorrect output rankings, sub-optimal model evaluations, and the amplification of undesirable behaviours in LLMs' alignment. This paper addresses the challenge of correcting such biases without additional data and training, introducing the concept of *Post-hoc Reward Calibration*. We first propose to use the local average reward to estimate the bias term and, thus, remove it to approximate the underlying true reward. We then extend the approach to a more general and robust form with the Locally Weighted Regression. Focusing on the prevalent length bias, we validate our proposed approaches across three experimental settings, demonstrating consistent improvements: (1) a 3.11 average performance gain across 33 reward models on the RewardBench dataset; (2) improved agreement of RM produced rankings with GPT-4 evaluations and human preferences based on the AlpacaEval benchmark; and (3) improved Length-Controlled win rate (Dubois et al., 2024) of the RLHF process in multiple LLM–RM combinations. According to our experiments, our method is computationally efficient and generalisable to other types of bias and RMs, offering a scalable and robust solution for mitigating biases in LLM alignment and evaluation.

## 1 INTRODUCTION

Reinforcement Learning from Human Feedback (RLHF) (Ouyang et al., 2022; Kaufmann et al., 2023) is driving the success of Large Language Models (LLMs) (OpenAI, 2023; Anil et al., 2023; Dubey et al., 2024). By integrating human preferences into the training loop, RLHF aligns models with desired behaviours and human values (Chakraborty et al., 2024), enabling LLMs to generate content that is conceptually appropriate (Sun et al., 2023), ethically sound (Hendrycks et al., 2021a; Dai et al., 2024), and reflecting nuanced human judgements (Wu et al., 2023). A key component in this process is the reward model (RM), which translates qualitative human feedback into a signal for optimising LLMs. The primary purpose of RM training is to approximate human preferences, allowing it to evaluate the content generated by LLMs by assigning scores. These scores (rewards) are then used to optimize the LLM, shaping its behaviour to better meet human expectations.

However, there is a risk that an RM may exploit spurious correlations in the training data that do not genuinely reflect the output quality (Stiennon et al., 2020; Gao et al., 2023; Singhal et al., 2023; Pang et al., 2023; Lambert & Calandra, 2023). For instance, an RM might favour outputs based on length (Park et al., 2024a) or style (Lambert et al., 2024; Dubois et al., 2023a), leading to biased assessments that do not accurately represent human preferences. The involvement of a biased RM in developing LLMs can have significant drawbacks. Firstly, it may lead to incorrect rankings of outputs, favouring content that aligns with learned biases over genuinely high-quality outputs. This issue is particularly problematic when RMs are used to rank different LLMs during evaluations, as a biased RM could produce misleading results, endorsing suboptimal models and impeding progress in model development. Moreover, when biased RMs are employed in RLHF, these biases can be further

amplified (Zheng et al., 2023b), with the LLM learning to exploit these biases rather than improving genuine task performance—a phenomenon known as *reward hacking*.

Recent advancements introduce various strategies to mitigate these biases and alleviate reward hacking during different stages of RLHF, from data handling (Park et al., 2024a) and reward model engineering (Coste et al., 2024) to the final reinforcement learning phase (Chen et al., 2024b). Most of these approaches, however, either require additional data collection, re-engineering of the RM, or modifications to reinforcement learning algorithms, which raises an intriguing question: is it possible to correct or mitigate biases in reward signals in a training-free manner? Specifically, given a set of scored outputs from the RM and a specific bias that the RM may exhibit, how can we calibrate these reward signals to reflect the true quality of the outputs more accurately?

This paper makes an initial attempt at answering this question by framing it as *Post-hoc Reward Calibration*. Specifically, for a particular characteristic of the output (*e.g.*, the length), we assume the biased reward from the RM can be decomposed into the sum of (1) the underlying calibrated reward and (2) a bias term that solely depends on that characteristic. Under certain assumptions, we propose to use the local average reward to provide an estimation for the bias term, which can then be removed, thereby approximating the true reward. We then extend it to be more general and robust using Locally Weighted Regression for bias estimation. Among different biases (Zhang et al., 2024b), a particularly prevalent one is the sequence length. Prior works showed that LLMs generate significantly longer responses after RLHF without improving the genuine quality of the content (Kabir et al., 2023; Wu et al., 2023; 2024b). Singhal et al. (2023) point out that even a purely length-based reward can reproduce most downstream RLHF improvements of existing RMs, highlighting that current RMs fail to convincingly outperform simple heuristics and may be strongly biased by length. The issue is also evident in powerful proprietary models like GPT-4, which is increasingly used as a judge, resulting in LLM evaluations that favour more verbose responses. Focusing on length bias, we validate our method in three settings and observe consistent improvements: (1) *Benchmark performance*: a **3.11** performance gain averaged over **33** RMs on the RewardBench; (2) *RM as LLMs evaluators*: based on AlpacaEval, we utilise **8** open-source RMs to rank **184** LLMs. After applying our calibration, the rankings correlate better with GPT-4 evaluations and human preferences. (3) *RM for LLMs' alignment*: we assess calibrated rewards for RLHF process. Testing with four LLM–RM configurations, we observe consistent improvements in AlpacaEval2 Length-controlled win rates.

In addition to being computationally efficient (*e.g.*, calibrating over 300k samples takes only 30 seconds with a single CPU), we empirically reveal that our method can generalise to other quantifiable biases, such as the presence of markdown-style features in the outputs, and to pairwise GPT4-as-Judge models. Importantly, the calibration effect is strongly correlated with the RM's preference for specific properties: for strongly biased RMs, the calibration yields greater improvements, while for weakly biased RMs, it only slightly alters its rewarding behaviour. These findings suggest that our proposed Post-hoc Reward Calibration method enhances the reliability of RMs and offers a practical, scalable solution for mitigating biases in RLHF. This approach could be instrumental in advancing RLHF methodologies and developing more robust and aligned LLMs.

## 2 RELATED WORKS

**Reward Models** Given a prompt $q$ and two corresponding responses $y_1$ and $y_2$, training an RM usually involves training a classifier to predict the human preference probability $p^*(y_1 \succ y_2|q)$ of two responses $y_1$ and $y_2$ with the Bradley-Terry (BT) model (Bradley & Terry, 1952):

$$p^*(y_1 \succ y_2|q) = \sigma(r^*(x_1) - r^*(x_2)) \tag{1}$$

where $x_i = (q, y_i)$ represents the prompt-response pair, $r^*$ is the latent true reward we cannot access, and $\sigma$ is the logistic function. Then, given a reward modelling dataset $\mathcal{D} = \{(q^i, y_1^i, y_2^i)\}_{i=1}^N = \{(x_1^i, x_2^i)\}_{i=1}^N$, where $y_1^i$ is annotated as a better response to prompt $q_i$ compared to $y_2^i$, training a RM $r_\theta$ is to minimise the negative log-likelihood for a binary classification problem:

$$\mathcal{L}(\theta) = -\mathrm{E}_{(x_1, x_2) \sim D}[\log \sigma(r_\theta(x_1) - r_\theta(x_2))] \tag{2}$$

In practice, RMs are often implemented by adding an extra linear layer to LLMs, which projects the representations of the last layers of LLMs to a scalar as the reward. There are also two other types of RMs: (1) DPO-based: LLMs optimised with Direct Preference Optimisation (DPO) (Rafailov

et al., 2023) could function as an implicit RM. (2) LLM-as-Judge: A generative LLM can also be prompted to function as an RM to generate feedback, usually consisting of a piece of chain-of-thought reasoning and a final score representing the overall quality (Zheng et al., 2023a; Kwon et al., 2023; Kim et al., 2024). In this paper, we validate the effectiveness of our methods across all three types of judge models. Specifically, we employ our method to calibrate BT-based and DPO-based RMs on the RewardBench dataset, and to calibrate the pairwise judge of GPT-4 with the AlpacaEval benchmark.

**Bias Mitigation for RLHF**   Recent works explore the bias mitigation, also termed as mitigate reward over-optimisation, for the RLHF process mainly from the perspective of (1) Data handling: Park et al. (2024a) identifies six types of inherent bias in various judge models and employ LLMs to generate the de-biasing dataset for further fine-tuning. Zhu et al. (2024) propose to iteratively replace hard labels in data with soft labels during RM training, while Rita et al. (2024) rely on human expert demonstration to recalibrate the reward objective optimised with Reinforcement Learning. (2) Reward Model Engineering: Coste et al. (2024); Eisenstein et al. (2023); Sun et al. (2024); Zhang et al. (2024a) implement the reward ensemble; Ramé et al. (2024) first fine-tune multiple RMs and then average them in the weight space; Dai et al. (2024) decouple the RM training for helpfulness and harmlessness and Chen et al. (2024b) disentangle the length signal and the quality signal into two different reward heads. Shen et al. (2023); Wang et al. (2024); Quan (2024) exploit the idea of Mixture-of-Experts (Qiu et al., 2024a;b) in RM training to capture different facets of human preference to avoid the potential conflict between them; (3) Reinforcement Learning methods: Another research line (Moskovitz et al., 2024; Sun et al., 2024; Zhou et al., 2024; Zhai et al., 2024) modifies the algorithm or optimisation objective during reinforcement learning with Park et al. (2024b); Zhu et al. (2024); Lu et al. (2024) specifically focusing on the length, one of the most consequential sources of bias (More discussions on length bias are in Appendix. A). Our work takes a different approach from previous methods by employing post-hoc reward calibration, *i.e.*, using only a batch of scored prompt-response examples to mitigate the RM's bias, without intervening the preference data collection, RM training, and the RLHF phase.

## 3   METHOD

### 3.1   PROBLEM STATEMENT: REWARD CALIBRATION

The reward calibration problem could be formalised as follows: given an input–output pair $x$ to be evaluated, the reward model $r_\theta$ assigns a scalar $r_\theta(x)$ to represent its quality. However, having been trained on the human preference dataset, the reward model $r_\theta$ may learn shortcuts to approximate the human preference and thus have biases regarding some characteristics of the input–output pair $x$. For example, it tends to assign higher rewards to outputs with longer length or containing knowledge commonly encountered in the real-world data (Park et al., 2024a). Supposing a measurable characteristic of interest $c$ is a function that maps the input–output pair $x$ to a scalar $c(x)$:

$$
\begin{aligned}
c: \quad \mathbb{X} \quad &\mapsto \quad \mathbb{R} \\
x \quad &\to \quad c(x)
\end{aligned}
$$

We assume that the biased reward from the reward model $r_\theta(x)$ could be decomposed into two terms, an underlying calibrated reward $r_\theta^*(x)$ and a bias $b_c^\theta(c(x))$ with regards characteristic $c$:

$$
r_\theta(x) = r_\theta^*(x) + b_c^\theta(c(x)) \tag{3}
$$

The RM is usually used to calculate the margin between two outputs $(x_1, x_2)$ to predict the human preference:

$$
\Delta_{r_\theta}(x_1, x_2) = r_\theta(x_1) - r_\theta(x_2) = r_\theta^*(x_1) - r_\theta^*(x_2) + b_c^\theta(c(x_1)) - b_c^\theta(c(x_2)) \tag{4}
$$

Thus, the reward calibration problem is to estimate the reward difference owing to the bias term and subtract it to recover the underlying true reward margin $\Delta_{r_\theta}^*(x_1, x_2) = r_\theta^*(x_1) - r_\theta^*(x_2)$.

**Post-hoc Reward Calibration**: The proposed reward calibration problem is applicable to various stages of RLHF. Depending on the focus, it can be tailored either toward collecting debiased preference data, refining the training of reward models, or even enhancing the alignment procedure by adjusting the alignment objectives. In contrast, this paper focuses on the *post-hoc* setting, performing reward calibration after the reward assignment without interfering with any data collection or RM

training stage. Formally, given $N$ already-rewarded data points $\{x_1^i, x_2^i, \Delta_{r_\theta}(x_1^i, x_2^i)\}_{i=1}^N$, we aim to find the calibrated reward margin $\hat{\Delta}_{r_\theta}^*(x_1^i, x_2^i)$ that is closer to the underlying oracle margin $\Delta_{r_\theta}^*(x_1^i, x_2^i)$ for every pair $(x_1^i, x_2^i)$.

**Assumptions**   We make the following hypothesis about the decomposition in Eq. 3.  (1) **Independence of the biased characteristic**: consider a general reward modelling dataset composed of prompts and corresponding responses pairs $\mathcal{D} = \{(q^i, y_1^i, y_2^i)\}_{i=1}^N = \{(x_1^i, x_2^i)\}_{i=1}^N$, the expectation of the underlying gold reward margin should be 0 and is independent of the biased characteristic $c$,

$$\mathbb{E}_{(x_1, x_2 \sim \mathcal{D})}\left[\Delta_{r_\theta}^*(x_1, x_2) \,\middle|\, c(x_1) = c_1, c(x_2) = c_2\right] = 0$$

(2) **Sufficient Density**: the function values of the characteristic function $c$ is assumed to be sufficiently dense in its range $c(\mathbb{X})$. (3) **Lipschitz Continuity**: the bias term $b_c^\theta$ is a *slow-varying function* of characteristic $c$, which means if pairs $x_1$ and $x_2$ are close to each other regarding the characteristic $c$, their corresponding bias term in Eq. 3 should be close as well. Mathematically, given $c(x_0)$ and a neighbourhood $\mathbb{V}(x_0) = \{c(x) \mid |c(x) - c(x_0)| < \epsilon\}$ of $x_0$, for all $c(x_1), c(x_2) \in \mathbb{V}$, there exists a constant $K$ such that $|b_c^\theta(c(x_1)) - b_c^\theta(c(x_1))| < K|c(x) - c(x_0)|$.

## 3.2   BIAS ESTIMATION

**Uniform Averaging Approach**   According to Eq. 4, the reward margin between $x_1$ and $x_2$ consists of (1) the underlying calibrated margin $\Delta_{r_\theta}^*(x_1, x_2)$ and (2) the biased "bonus reward" that solely depend on the characteristic $c(x_1)$ and $c(x_2)$. Intuitively, to estimate $b_c^\theta(c(x_1)) - b_c^\theta(c(x_2))$, we ask:

**Given the characteristic measures $c(x_1)$ and $c(x_2)$ for an arbitrary pair $(x_1, x_2)$, what would be their reward margin?**

A straightforward estimation for this question could be

$$\mathbb{E}[r_\theta(x) \mid c(x) = c(x_1)] - \mathbb{E}[r_\theta(x) \mid c(x) = c(x_2)]. \tag{5}$$

Under the assumption of Lipschitz continuity and sufficient density, we can obtain a simple calibrated reward margin described in Eq. 6:

$$\hat{\Delta}_{r_\theta}^*(x_1, x_2) = \Delta_{r_\theta}(x_1, x_2) - \left(\mathbb{E}[r_\theta(x) \mid |c(x) - c(x_1)| < d] - \mathbb{E}[r_\theta(x) \mid |c(x) - c(x_2)| < d]\right) \tag{6}$$

where $d$ is a threshold distance that governs the neighbourhood size around $x_1$ and $x_2$.

**Locally-Weighted-Regression Method**   How do we choose the threshold distance $d$ in Eq. 6? (1) It must be sufficiently small to ensure that the bias term can be approximated as a constant function within the local neighbourhood, enabling the application of Eq.6; and (2) it must also be large enough to meet the sufficient density assumption, ensuring sufficient data points used for estimation. Satisfying these conflicting requirements may necessitate thorough hyper-parameter tuning of $d$ to employ the intuitive approach in practice. Furthermore, the density may not be uniform in practice, so that the optimal threshold distance $d$ may vary for different data points $c(x)$. Therefore, a desirable approach should be able to estimate the expectations in Eq. 5 in a more general, robust, and practical way by (1) relaxing the constant-function assumption and (2) mitigating the sensitivity to the choice of neighbourhood size, possibly by incorporating certain adaptive mechanisms to optimise the neighbourhood size or the contribution for each datapoint dynamically. Both desiderata motivate us to employ Locally Weighted Regression (LWR) towards a more robust estimation.

Specifically, given a point of interest (*e.g.*, a characteristic value $c(x)$ in our case), the Locally Weighted Regression (LWR) method first uses a bandwidth parameter $f$ to define the fraction of the dataset utilised for regression. Then the method assigns weights to neighbouring data points, giving higher weights to points closer to the target and lower weights to those farther away. This approach ensures that nearby points have a greater influence on the regression. LWR then fits a weighted linear regression to model the local behaviour of the target function, effectively approximating the weighted average within the specified local context.

LWR addresses the desiderata by offering a more adaptable approach to estimate the Eq. 5. It relaxes the constant-function assumption by introducing the linear approximation to capture the

subtle local variations in the data. This is reasonable because the bias terms are slow-varying functions and can be linearly approximated using Taylor's expansion. Furthermore, LWR mitigates sensitivity to neighbourhood size by using a distance-based weighting mechanism, which assigns greater importance to closer data points. This adaptive weighting scheme dynamically adjusts the influence of each data point, reducing reliance on a fixed neighbourhood size and enhancing the robustness and flexibility of the estimation process. Formally, given a dataset consisting of the characteristic $c$ in outputs and corresponding rewards $D_c = \{(c(x_j), r_\theta(x_j))\}_{j=1}^N$ and a bandwidth $f$, our general method could be formulated as:

$$\hat{\Delta}^*_{r_\theta}(x_1, x_2) = \Delta_{r_\theta}(x_1, x_2) - (\hat{r}_\theta(c(x_1)) - \hat{r}_\theta(c(x_2))) \tag{7}$$

where $\hat{r}_\theta(c(x_1))$ and $\hat{r}_\theta(c(x_2))$ are the LWR prediction for the characteristic $c(x_1)$ and $c(x_2)$, respectively, given the dataset $D_c$. To determine the preference, we use the sign of reward margin between $x_1$ and $x_2$: $\mathrm{sgn}(\hat{\Delta}^*_{r_\theta}(x_1, x_2))$. In practice, we employ the Robust Locally Estimated Scatterplot Smoothing (LOWESS), a robust version of LWR. Based on LWR, it defines different weights $\delta_j$ for the point $(c(x_j), r_\theta(x_j))$ according to the size of residual $r_\theta(x_j) - \hat{r}_\theta(c(x_j))$. It assigns smaller weights to data points with large residuals and large weights to those with small residuals. New fitted values are computed using LWR again but with $w_j$ replaced by $\delta_j w_j$. The computation of new weights and fitted values is repeated several times to converge. The full algorithm of using LOWESS for reward calibration is described with Appendix B Algorithm 1. We directly employ off-the-shelf library `statsmodels.api` and pass the dataset $D_c$ to it for our implementation.

**Length Bias Estimation** This paper prioritises the prevalent length bias presented in the reward model. Thus, the characteristic of interest here is the character length of the output, *i.e.*, $c(x) = |x|$. The dataset used for regression consists of the length of the output and its corresponding rewards.

## 4 EXPERIMENTS

One key advantage of our proposed *Post-hoc* reward calibration is that it does not need any extra data or training and thus can be seamlessly applied to various scenarios. So, we test the proposed method in three settings: (1) *RM Benchmark performance*: we test the calibrated RM's performance on the RewardBench dataset (Lambert et al., 2024). (2) *RM As LLMs Evaluator*: Another critical application for RM is to serve as a judge to replace human evaluation when evaluating LLMs. We thus also explore whether the calibrated RM could provide a better evaluation. Based on the AlpacaEval benchmark (Dubois et al., 2023a), we employ RMs to label the output of different LLMs from AlpacaEval leaderboard and produce a ranking of LLMs. Then, we measure the correlation between the RM's ranking and GPT-4 ranking or human preference ranking. (3) *RM for LLMs' alignment*: Finally, as discussed in our introduction, employing a biased RM risks amplifying the potential bias and thus leads to unexpected behaviours. So, we explore how our proposed calibration method affects the LLM alignment results. Specifically, we use the calibrated rewards to label the preference data, and then conduct Direct Preference Optimization (Rafailov et al., 2023) with the labelled preference pair. We evaluate the aligned LLM's performance using AlpacaEval2 and eight popular benchmarks. In all three settings, we are given a group of data points $D_{|x|} = \{x_i, r_\theta(x_i)\}$ for calibration. Because LOWESS requires calculating the residual for every fitted data point, we employ the entire $D_{|x|}$ for regression. Therefore, for RewardBench, there are 2,985 test data points, so we employ all 5,970 data points to perform the regression (every test data is a pair of responses for one prompt). Regarding AlpacaEval, there are 184 LLMs and 805 instructions for each LLM, resulting in 151k samples used for calibration. For LLMs' alignment, five responses are generated from the LLM for about 60k instructions, leading to about 300k samples used for bias estimation. The proposed method is computation-efficient, and calibrating 300k samples only takes 30 seconds using a single CPU.

**Compared Algorithms** For all three experimental settings, we try to compare the following algorithms: (1) `Original reward`; (2) `Length penalty`: a broadly adopted approach to mitigate the length bias in reward models. It adds a penalty term to the original reward (Singhal et al., 2023; Park et al., 2024b; Liu et al., 2024a; Dong et al., 2024), formalised as $\hat{r}^*(x) = r_\theta(x) - \alpha \times |x|$. Following Dong et al. (2024), we use the character length of each output and set $\alpha = 0.001$. (3) `RC-Mean`: our proposed uniform averaging calibration method described in Eq. 6. (4) `RC-LWR`: our proposed Locally Weighted Regression calibration method shown in Eq. 7. (5) `RC-LWR-Penalty`:

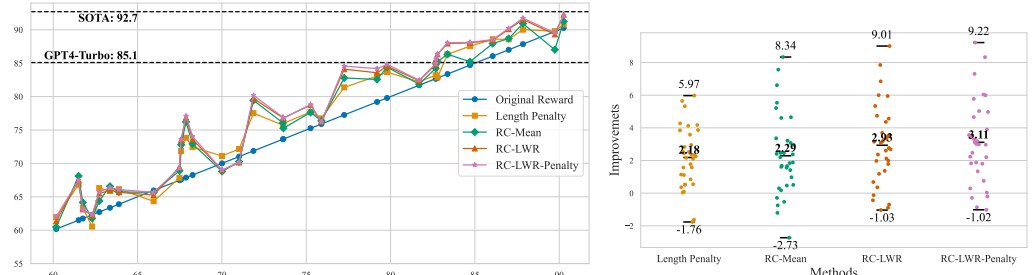

Figure 1: Results of different calibration algorithms on the RewardBench benchmark are shown in two charts. **Left**: the line chart demonstrates the RewardBench score before calibration (x-axis) and after calibration (y-axis) of different algorithms for different models. **Right**: the scatter plot highlights the performance gains achieved by different calibration methods, annotated with the maximum, average, and minimum values.

to see whether our method could be applied on top of the penalty-based methods, we first apply the length penalty to the original reward and then use the `RC-LWR`. All hyper-parameters related to `RC-Mean`, `RC-LWR` and `RC-LWR-Penalty` are presented in Appendix C.

## 4.1 LENGTH CALIBRATED REWARDS ON REWARDBENCH

**Setting and Metrics** Each data point in RewardBench comprises one prompt $q$ and two LLMs-generated responses $y_w$ and $y_l$. The final score is calculated as the averaged accuracy of reward models in identifying $y_w$ over four subsets: Chat, Chat Hard, Safety, and Reasoning. We mainly investigate **33** BT-based RMs whose rewarding scores are officially available[1] and whose accuracy is above 50% (random guess) from the RewardBench Leaderboard as of Aug. 2024.

**Results** The overall results for different calibration methods are illustrated in Fig. 1. The line chart on the left showcases the calibrated performance (y-axis) w.r.t original performance (x-axis) of different calibration algorithms for different RMs. The scatter plot on the right highlights the performance gains for different calibration methods. Our empirical findings include: (1) The simple coarse-grained `Length Penalty` method already leads to a notable improvement (orange line v. blue line), to some extent indicating the prevalence and significance of the RM's length bias. (2) The reward calibration techniques we introduce yield improvements for nearly all RMs. Among these techniques, the `RC-LWR-Penalty` achieves a 3.11 performance gain averaged over all 33 models. Though it could lead to performance degradation (5 models for `RC-LWR` and 4 models for `RC-LWR-Penalty`), the degradation is very slight and the worst case is only about -1 point for `RC-LWR` and `RC-LWR-Penalty`. (3) More importantly, besides improving reward models with moderate performance, the proposed calibration method could also boost the performance of strong reward models. For example, after `RC-LWR-Penalty` calibration, nine reward models surpass the accuracy of `GPT4-Turbo`, and the best calibrated model even approaches the latest state-of-the-art (SOTA) LLM-as-Judge models. We also calibrate DPO-based reward models with our proposed method and report results in the Appendix D.

## 4.2 LENGTH CALIBRATED REWARD AS LLMS EVALUATORS

**Settings** The typical LLM evaluation pipeline first constructs a prompt dataset, samples responses from different LLMs, and then calculates the win rate between different models. Based on this pipeline, AlpacaEval (AE1) (Dubois et al., 2023b), with `GPT4-1106-Preview` as the evaluator, fixes a baseline LLM and uses the win rates against the same baseline LLM to produce the ranking. To calibrate the length bias of GPT4 evaluation, AlpacaEval2 (AE2) (Dubois et al., 2024) introduced length-controlled regression to calibrate the win rate of AE1 and became a popular generative LLM leaderboard. Based on the AE1 prompt dataset, this section tests whether the open-source BT-based RMs could provide a more reliable evaluation for LLMs after calibration. Because the length-controlled technique proposed in AE2 can apply to BT-based RMs, we include it in our baselines, denoted as `LC`. We also apply our method to calibrate AE1 to further compare with `LC`. More details about applying RM for AE and calibrating the produced ranking are in Appendix C.

**Metrics** Following Dubois et al. (2024), we look at the following metrics to see whether the calibrated rewards could provide a more reliable evaluation: (1) *Gameability*: this measures whether

---

[1] 

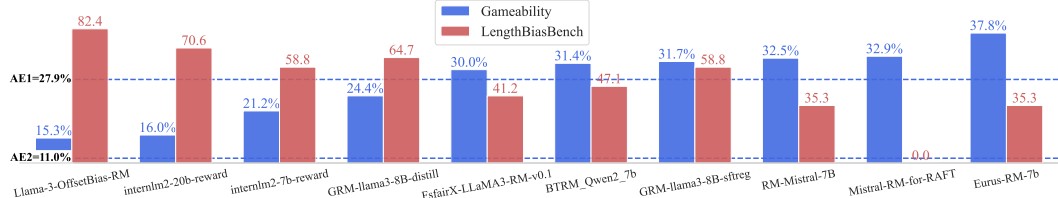

Figure 2: The Gameability ↓ and LengthBiasBench accuracy ↑ for selected BT-based reward models. The lower the Gamebility, the less sensible the win rate is to different prompt strategies. The higher the accuracy on the LengthBiasBench, the less preference the model has for output length. The Gameability of AlpacaEval1 (AE) and AlpaceEval2 (AE2) win rates are 11.05% and 27.9%, respectively, presented with the dashed line.

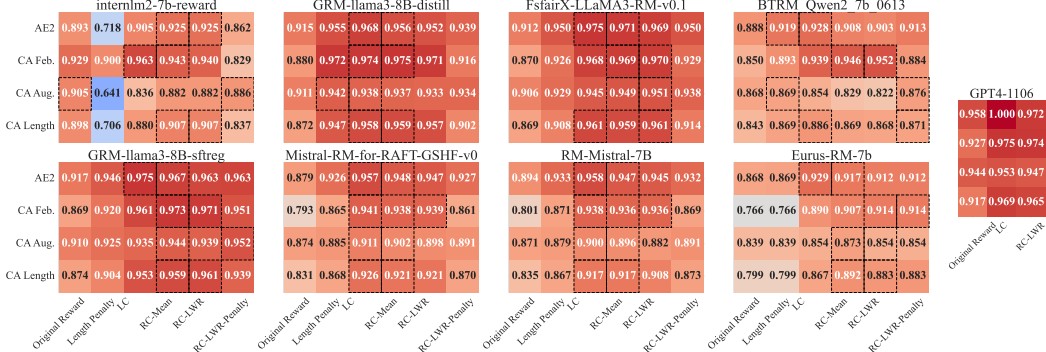

Figure 3: The heatmap demonstrates the Spearman correlation (↑) between reward-models-produced rankings with the AlpacaEval2 (AE2) and ChatbotArena (CA). We also report the calibration results for `GPT4-1106` on the right, with the `Original Reward` indicating AlpacaEval1, `LC` representing AE2, and `RC-LWR` referring to rankings calibrated with our proposed `RC-LWR` method. The x-axis represents different calibration methods. The two highest correlations in each row are marked with a dashed rectangle. We select three versions of CA: (i) CA Feb., from the AE2 paper, to validate our reproduction of the original results; (ii) CA Aug., the latest ranking which contains more LLMs; and (iii) CA Length, which is CA Aug. disentangled with respect to the response length.

the win rate given by the evaluator is significantly affected by simply promoting the same model to be more concise or verbose. Like Dubois et al. (2024), five LLMs are selected to be prompted to generate the response normally, verbosely, and concisely. Then, the gameability is defined as the normalised variance between those three win rates and averaged across these five models. Higher gameability indicates that the evaluator is more sensitive to the output's length and could help us identify evaluators with length bias. (2) *Spearman correlation* with existing popular LLM leaderboard, including AE2 and ChatbotArena (CA). We employ three versions of (CA): (i) CA Feb., used by Dubois et al. (2024), to reproduce their results; (ii) CA Aug., the latest version that includes more LLMs; and (iii) CA. Length, namely CA Aug. calibrated with respect to length as proposed by Chiang et al. (2024).

**Identifying RMs with strong length bias**  Before calibration, we first evaluate the top-10 BT-based RMs from RewardBench leaderboard as of Aug. 2024 using the Gameability and the length bias subset of the EvalBiasBench dataset (Park et al., 2024a). The latter contains 17 test examples and is denoted as LengthBiasBench. We aim to identify RMs demonstrating severe length bias, *i.e.*, high Gameability and low accuracy on LengthBiasBench. Their results are shown in Fig. 2. We observe the prevalence of length bias for BT-based RMs. 8 of 10 models exhibit high Gameability and poor performance on LengthBiasBench. The two metrics demonstrate consistency, *i.e.*, the model with high Gameability usually achieves low accuracy on LengthBiasBench, achieving `-0.879` Spearman correlation score. As `Llama-3-OffsetBias-RM` and `internlm2-20b-reward` do not demonstrate severe length bias, we primarily report results for other 8 models.

**Results**  The Spearman correlations of RM-produced rankings with the AE2 and the ChatbotArenas are presented in the Fig. 3. Our findings include: (1) Overall, `RC-Mean` and `RC-LWR` provide effective calibrations, leading to a higher correlation with AE2 and ChatbotArena, achieving comparable calibration effects with `LC` on all eight RMs. Note that `LC` is specifically designed for the AE leaderboard and can not be straightforwardly employed for reward calibration because it requires the LLM ID and instruction ID for regression, which are not available for general reward calibration. (2) Compared with AE2, `RC-LWR` calibrated AE achieves comparable Spearman correlations with Chatbot Arena, suggesting that our method can generalise to LLM-as-Judge results. (3) The cali-

Table 1: The Gameability ↓ of different calibrations methods across various reward models on AlpacaEval

| Model | Length Penalty | LC | RC-Mean | RC-LWR | RC-LWR-Penalty |
|---|---|---|---|---|---|
| internlm2-7b-reward | 13.6% | 10.6% | 9.4% | **9.0%** | 25.6% |
| GRM-llama3-8B-distill | 12.1% | 9.5% | **8.8%** | 9.0% | 16.7% |
| FsfairX-LLama3-RM-v0.1 | 20.2% | 12.7% | 10.8% | **10.1%** | 19.0% |
| BTRM-Qwen2_7b | 18.5% | 13.8% | **10.5%** | 10.6% | 17.5% |
| GRM-llama3-8B-sftreg | 23.4% | 13.7% | **9.7%** | 10.0% | 14.8% |
| RM-Mistral-7B | 20.5% | 14.2% | 10.8% | **10.5%** | 19.8% |
| Mistral-RM-for-RAFT | 22.0% | 13.5% | 9.9% | **9.6%** | 19.5% |
| Eurus-RM-7b | 37.8% | 18.3% | 12.2% | **11.8%** | **11.8%** |

brated `GRM-llama3-8B-distill` achieves `0.975` with ChatbotArena Feb., and the calibrated `FsfairX-LlaMA3-RM-v0.1` achieves `0.975` with AE2 and `0.951` with ChatbotArena Aug., demonstrating the potential of calibrated open-source RM to provide reliable LLM evaluations. (4) The `Length Penalty` brings improvements in most cases. But it can also lead to degeneration (`internlm2-7b-reward`) or can be not effective at all (`Eurus-RM-7b`). This is because different RMs have different reward scales, making it harder to employ the same penalty weight for different RMs. On the contrary, our proposed `RC-LWR` is more robust to different hyper-parameter choices, rendering it easier to use in practice. (5) Unlike the results on RewardBench, the `Length Penalty` and `RC-LWR-Penalty` are less effective in this setting, likely due to differences in length margin ( denoted as $||x_1| - |x_2||$) distributions of two dataset. For instance, in the RewardBench, 60% of data points have a length margin below 600, while in the AlpacaEval setting, this figure is less than 40%. Consequently, the penalty term has a more significant impact on the AlpacaEval calibration. More relevant analyses are shown in the Appendix D. This underscores the practicality of our method over the penalty-based approach, as the latter may require extensive hyperparameter tuning for penalty weight. Such tuning is influenced not only by the RM's reward scale but also by the output's length distribution. (6) The rankings calibrated by `LC`, `RC-Mean`, `RC-LWR` are usually more aligned with CA Length compared with CA Aug., further validating the effectiveness of our proposed methods. (7) As shown in Tab. 1, `RC-Mean` and `RC-LWR` effectively reduce the Gameability for various reward RMs, demonstrating the calibrated rewards are less sensitive to different prompting techniques and thus more reflective of the response quality.

### 4.3 LENGTH CALIBRATED REWARDS FOR LLMS' ALIGNMENT

**Setting** In this setting, we investigate whether the calibrated rewards could serve as a better training signal for LLMs' alignment. Because the post-hoc reward calibration assumes a scored dataset used for calibration, which could be seamlessly integrated with the offline RL, we choose to first employ the RMs to score the on policy dataset and then perform DPO. Our experiment pipeline could be described as follows: for each prompt $q$ in a prompt dataset, multiple responses $y_i$ are sampled from an LLM, then scored as $r_i$ by an RM. Among these responses, the response with the highest or lowest score is selected to create a preference pair $(p, y_w, y_l)$ for Direct Preference Optimisation (DPO) tuning. Specifically, to validate the effectiveness of our method, we choose two instruct LLMs: `Meta-Llama-3-8B-Instruct` (AI@Meta, 2024) and `gemma2-9b-it` (Team, 2024) and two reward models: `FsfairX-LLaMA3-RM-v0.1` (Dong et al., 2023) and `GRM-llama3-8B-sftreg` (Yang et al., 2024), resulting in 4 LLM-RM configurations: (1) `Llama-3-8B-Fsfairx`; (2) `Llama-3-8B-GRM`; (3) `gemma2-9b-Fsfairx`; and (4) `gemma2-9b-GRM`. For each LLM–RM combination, we compare the `Original Reward`, `Length Penalty`, and `RC-LWR`. We directly utilise the on-policy generations[2] of two selected LLMs generated by Meng et al. (2024). They employ the 60k instructions from the UltraFeedback (Cui et al., 2024) dataset and sample 5 responses for each instruction from the LLM. Then, we use those two RMs to score the dataset and to conduct DPO tuning. All hyper-parameters related to DPO tuning are set following Meng et al. (2024).

**Metrics** We evaluate the LLM with the Length-Controlled win rate from the AE2 for open-ended generation. We also test the LLM on eight popular benchmarks: MMLU (Hendrycks et al., 2021b), GSM8K (Cobbe et al., 2021), ARC-challenge (Clark et al., 2018), HellaSwag (Zellers et al., 2019), Winogrande (Sakaguchi et al., 2021), IFEval (Zhou et al., 2023), GPQA (Rein et al., 2023), and BBH (Suzgun et al., 2023) to see how the proposed calibration affects the benchmark performance. We employ the OpenCompass (Contributors, 2023) for benchmark evaluation and report the average score of these benchmarks. All evaluation configurations are set as a default in OpenCompass.

---

[2]`huggingface.co/datasets/princeton-nlp/llama3-ultrafeedback-armorm` and `huggingface.co/datasets/princeton-nlp/gemma2-ultrafeedback-armorm`

Table 2: Results of using length-calibrated rewards for LLMs' alignment. Calibration methods we compare include `Original Reward (OR)`, `Length Penalty (LP)`, and `RC LWR`. We apply these methods in four different LLM–RM configurations. We report the Length-Controlled win rate, the average character length on the AlpacaEval, and average performance on eight benchmarks. The Table illustrates that `RC-LWR` calibration achieves up to 10% LC win rate improvement and alleviates the performance drop on benchmarks.

| Algorithm | no DPO | OR | LP | RC LWR | no DPO | OR | LP | RC LWR | no DPO | OR | LP | RC LWR |
|---|---|---|---|---|---|---|---|---|---|---|---|---|
| | Length-Controlled Win Rate | | | | Avg Character Length in Alpaca | | | | Avg Performance on 8 Benchmarks | | | |
| `Llama-3-8B-Fsfairx` | 23.91 | 41.82 | 49.98 | **51.37** | 1967 | **2404** | 1989 | 1867 | 62.69 | 60.45 | 61.75 | **61.94** |
| `Llama-3-8B-GRM` | 23.91 | 41.00 | 48.41 | **50.49** | 1967 | **2438** | 1996 | 1858 | 62.69 | 60.94 | **62.11** | 61.83 |
| `gemma2-9b-Fsfairx` | 46.96 | 62.79 | 67.46 | **69.54** | 1521 | **2365** | 1491 | 1798 | 63.69 | 58.88 | 60.52 | **60.79** |
| `gemma2-9b-Grm` | 46.96 | 63.21 | 66.28 | **70.45** | 1521 | **2106** | 1426 | 1780 | 63.69 | 55.28 | 57.28 | **58.31** |

**Results**    Results of employing calibrated rewards for LLMs' alignment are demonstrated in Tab. 2, where the left and the middle focus on the LC win rate and the right on benchmark performance. We observe that: (1) Similar to the experimental results with the RewardBench, the `Length Penalty` serves as a simple yet effective baseline, highlighting the significant issue of length exploitation in LLM alignment (Meng et al., 2024). (2) Without extra training and data annotation, the `RC-LWR` outperforms the `Original Reward` baseline on AE2 by about `9.5` for `Meta-Llama-3-8B-Instruct`-based DPO and about `7` points for `gemma2-9b-it`. It effectively improves the output quality while preserving the length. (3) The DPO tuning leads to performance drops on our eight benchmarks. The drop is more significant for `gemma2-9b-it`-based configurations (*e.g.*, for `gemma2-9b-fsfairx`, the performance drops from 63.69 to 55.28). This may be because a larger learning rate of 5e-7 is set for gemma2-based experiments compared to 3e-7 for Llama3-based experiments following Meng et al. (2024). However, with the same hyper-parameters, both `Length Penalty` and `RC-LWR` methods could mitigate the performance drop, indicating that the length-regularised alignment not only controls the output length but also reserves the LLM's benchmark performance during DPO alignment.

## 5 ANALYSIS

| | Original Reward | Length Penalty | RC-Mean | RC-LWR | RC-LWR-Penalty |
|---|---|---|---|---|---|
| $|\rho|_{\text{avg}}$ | 0.2930±0.1836 | 0.2092±0.1706 | 0.0390±0.0369 | 0.0233±0.0223 | 0.0229±0.0223 |

Table 3: The average and variance of absolute correlation score for different methods on RewardBench.

**Does the calibration mitigate the length preference of RMs?**    One desideratum for an ideal calibration method is that the rewards should be weakly correlated with the bias characteristics of interest, thus cancelling the RM's preference for specific characteristics and preventing reward hacking in its downstream applications. Therefore, we calculate the average and variance of the **absolute** Spearman correlation score over RMs for different methods on the RewardBench dataset and show them in Tab. 3. Our observation includes: (1) Length bias is significant for the broad spectrum of RMs (`0.293` for original reward). (2) The `Length Penalty` cannot effectively mitigate the RM's preference for length. The added penalty term renders RM to prefer shorter outputs. (3) In contrast to `Length Penalty` method, our proposed methods could effectively calibrate the RM's preference on output length, *i.e.*, the correlation score between output length and reward is approximately 0 for all RMs after calibration.

**Calibration behaviours on RMs with different length preferences.**    If an RM exhibits a weak preference for the characteristic of interest, an optimal calibration should not significantly alter its rewarding behaviour and vice versa. Rewarding accuracy alone may not effectively capture this aspect, as a calibration might result in an equal number of correct and incorrect predictions, thereby leaving the overall accuracy unchanged. We thus count how many preference pairs are reversed by the calibration on the RewardBench dataset. We report the fraction of it over the entire dataset. The results are shown in the Fig. 4. Our observations are as follows: (1) `Length Penalty` demonstrates the moderate correlation between RM length bias and the preference overturned and is unstable. It does not calibrate any pairs for some RMs with strong positive length preferences. For RMs with a strong negative preference for length, it is ineffective the RM already prefers shorter outputs in general. (2) Both `RC-Mean` and `RC-LWR` methods exhibit a strong correlation: the weaker the bias of the

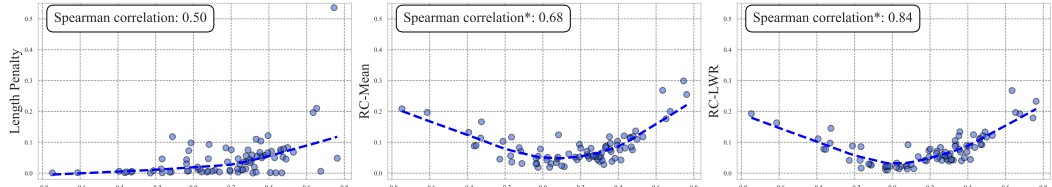

Figure 4: The number of preferences reversed by the calibration (y-axis) for RMs with different length preferences (x-axis), measured by the Spearman correlation with length. To determine the correlation between x and y, we further report the Spearman Correlation between them. For `RC-Mean` and `RC-LWR`, as the plot is approximately symmetric, we report the correlation between absolute values of x and y.

RM, the fewer pairs are reversed by the calibration method, indicating that our calibration method will not severely affect the non-biased RM. The desirable feature may enhance the practicability of the proposed method.

**Generalisation to other bias types: markdown features.** To validate that our proposed method could generalise to other bias, we apply it to calibrating the bias towards the markdown listing features in the output, motivated by the empirical findings of Dubois et al. (2023a). Observing that LLMs often utilise markdown style lists with bold subheadings as the output format, we define the characteristic function $c(x)$ as the number of headers, lists, and bold strings following Chiang et al. (2024). The calibration results for 33 BT-based RMs are shown in Fig. 5. We find that calibrating the number of lists in the output also yields improvement on the RewardBench,

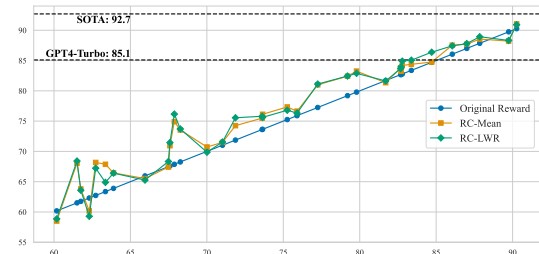

Figure 5: RewardBench score before (x-axis) and after (y-axis) markdown-feature-calibration.

with a `1.81` averaged performance gain for `RC-Mean` and `1.86` for `RC-LWR`. Furthermore, similar to length bias calibration, improvements are observed across nearly all RMs, and the originally good RMs get slightly stronger after calibration. The gains are not as big as the gains for calibrating length bias, which demonstrates, to some extent, that the length bias is more prevalent.

**Further Analysis** In Appendix D, we show that the proposed method is robust to different hyperparameter choices of LWR (Fig. 7). For the cases when the independence assumption does not hold, we introduce a calibration constant before LWR prediction to determine how much bias to subtract, thus smoothly controlling the calibration effect (Fig. 8 and Tab. 4). We also demonstrate that RMs with more substantial bias benefit more from our approach (Fig. 10). We further showcase the potential of the proposed method to calibrate bias from multiple characteristics simultaneously (Fig. 11). Moreover, a stress test evaluating the data efficiency of the method reveals that our method is not significantly constrained by the size of the datasets used for the calibration (Fig. 13).

## 6 CONCLUSION

This paper introduces a post-hoc reward calibration method to mitigate biases in RMs. The proposed approach effectively reduces the impact of prevalent biases, such as length bias, and has been validated across multiple experimental settings. The results consistently show substantial performance gains. Other empirical findings confirm that our method satisfies several critical desiderata for optimal calibration: (1) Our approach consistently mitigates both length and markdown biases in RMs, and can successfully calibrate LLM-as-Judge rewards, underscoring its generality. (2) The method requires no additional data annotation or RM retraining and is computationally efficient. (3) The calibrated rewards exhibit weak correlations with the characteristics of interest, thereby preventing reward hacking and improving the overall quality of the reward signals. (4) For non-biased RMs, our method minimally alters the reward behaviour, whereas it substantially improves RMs with stronger biases, making the calibration impact proportional to the bias level. (5) The method is robust to hyperparameter choices, and a calibration constant can be introduced to preserve some desirable bias towards the specific feature, enhancing its practical usability. (6) The approach is not constrained by dataset size, as even a few hundred data points are sufficient to achieve effective calibration. In conclusion, our approach offers a scalable, efficient, and generalisable solution for addressing biases in RMs, ensuring better alignment between LLMs and human preferences and LLMs evaluation.

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

## A   LENGTH BIAS

Length bias emerged as a significant issue in Natural Language Processing (NLP), affecting a wide range of tasks and applications, including machine translation (Zhang et al., 2023; Murray & Chiang, 2018) and text summarization Guo & V∫osoughi (2023); Steen & Markert (2024). This bias occurs when models disproportionately prefer outputs of a certain length, which stands as one of the most consequential sources of bias in NLP.

In LLM era, length bias (also termed as the *verbosity issue*) continued to be a pressing concern. One key process to develop such LLMs is RLHF, which employ human preference data to fine-tune LLMs (directly or indirectly) to generate more human-aligned outpus but also may unintentionally amplify the length bias in the preference dataset. Therefore, mitigating length bias become innvreasingly important, drawing significant attention. Prior works include: (1) **works focusing on RMs training**: Chen et al. (2024b) propose to jointly train two linear reward heads on top of the same model, one trained to correlate with length and the other trained to decorrelate with length, their experiments demonstrate that the approach eliminate the Pearson correlation to 0 (same as our approach); Similarly, Shen et al. (2023) jointly optimise a main expert RM and a length-bias only expert RM, and exclusively use the main expert in the RL phase; Hou et al. (2024) devise Bucket-Based Length Balancing, that they divide the preference data used to train RM in to different bucket based on length difference, and within each bucket, they balance the number of examples where the better/worse response is more lengthy; (2) **works on focusing on LLM alignment**: Meng et al. (2024) leverages the length-normalised reward to prevent model from generating longer but low-quality responses; Lu et al. (2024) modify the DPO objective by down-sampling same amount of tokens from the chosen and rejected responses when calculating the implicit reward, similar with Liu et al. (2024b); Chen et al. (2024a); Liu et al. (2024a); Park et al. (2024b) add an extra penalty term to the implicit reward to penalise the verbosity; Wu et al. (2024a) set a threshold for top-tier response and select the shortest one as the best response when constructing the preferenc data; and (3) **works focusing on the evaluation**: Based on AlpacaEval benchmark, Dubois et al. (2024) propose use LLM ID and instruction ID to do a regression to estimate the bias; Chiang et al. (2024) employ the same method for Chatbot Arena.

All these previous studies underscore the prevalence and significance of length bias across different subfields of LLMs, highlighting the pressing need for a unified approach to address this issue comprehensively. While our method was not specifically designed to target length bias, it has been empirically validated to effectively mitigate this problem in various contexts, demonstrating its broader applicability.

## B    FULL ALGORITHMS

The algorithm describing employing the LOWESS method for length bias calibration is shown in Algorithm 1. The main idea is to first fit a locally weighted regression and then adjust the weight based on the residual between regression prediction and the ground truth label. LOWESS is a non-parametric regression technique typically used for smoothing a scatterplot. It is particularly useful when no specific functional form is assumed. Key uses of LOWESS include: (1) **Data Smoothing**: LOWESS is used to smooth out noise in data, providing a clearer trend by creating a smooth curve that captures the underlying pattern. (2) **Data Analysis**: By overlaying a LOWESS curve on a scatterplot, one can explore the potential relationship between variables without relying on predefined functions, and it's particularly useful when the relationship between variables is complex and non-linear, as it adapts to the shape of the data. Based on the typical usage of LOWESS, in this paper, we employ LOWESS to investigate the potential relationship between rewards from a biased RM and a characteristic in the output. By assuming that they are actually independent, we regard the prediction from the LOWESS as the bias term and remove it to recover the underlying true reward.

---

**Algorithm 1** LOWESS for Length Bias Calibration

---

1: **Input:** Output length and corresponding rewards $(|x_i|, r_\theta(x_i))$ for $i = 1, , 2, \ldots, n$, the bandwidth $f$ to demterming the fraction of dataset used for regression, the number of iterations $k$
2: **Output:** Calibrated rewards $\hat{r}_\theta^*(x_i)$
3: **for** each $x_i$ **do**
4:     Compute the distances $d_{ij} = ||x_i| - |x_j||$ for all $j = 1, 2, \ldots, n$
5:     Determine the $d_i$ as the distance to the $q$-th nearest neighbor, where $q = \lceil fn \rceil$
6:     Compute the tricube weights $w_{ij} = \left(1 - \left(\frac{d_{ij}}{d_i}\right)^3\right)^3$ for $d_{ij} \leq d_i$
7:     Fit a weighted linear regression near $x_i$: $\arg\min_{\beta_0, \beta_1} \sum_j w_{ij}(r_\theta(x_j) - \beta_0 - \beta_1|x_j|)^2$
8:     Compute the fitted value $\hat{r}_\theta(|x_i|) = \hat{\beta}_0 + \hat{\beta}_1|x_i|$
9: **end for**
10: **for** $t = 1$ to $k$ **do**                                      ▷ Robustifying iterations
11:     Compute the absolute residuals $\Delta_i = |r_\theta(x_i) - \hat{r}_\theta(|x_i|)|$
12:     Compute the median of absolute residuals: $s = \text{median}(\Delta_i)$
13:     Calculate robustness weights $\delta_i = B\left(\frac{\Delta_i}{6s}\right)$, where $B(u) = \max(0, 1 - u^2)^2$
14:     **for** each $x_i$ **do**
15:         Combine weights: $w_{ij}^{\text{new}} = \delta_j w_{ij}$
16:         Refit the weighted linear regression: $\arg\min_{\beta_0, \beta_1} \sum_j w_{ij}^{\text{new}}(r_\theta(x_j) - \beta_0 - \beta_1|x_j|)^2$
17:         Compute the new fitted value $\hat{r}_\theta(|x_i|) = \hat{\beta}_0^t + \hat{\beta}_1^t x_i$
18:         Compute the calibrated rewards $\hat{r}_\theta^*(x_i) = r_\theta(x_i) - \hat{r}_\theta(|x_i|)$
19:     **end for**
20: **end for**
21: **return** Calibrated rewards $\hat{r}_\theta^*(x_i)$

---

## C    EXPERIMENTAL DETAILS

**Experiments with AlpacaEval**    AlpacaEval 2.0 is a popular LLM benchmark ranking over 200 models. The official leaderboard employs `GPT4-1106-preview` as judge. In our setting, we primarily use open-source BT-based RMs to score the LLM's output. Given a prompt $q$ and corresponding output $y$ from the evaluated LLM and output $y_b$ from the baseline LLM, the RM scores them as $r$ and $r_b$, respectively; then the Bradley-Terry model yields the preference probability $p(y \succ y_b|q) = \sigma(r - r_b)$. We then compute the average of the perference probability $p(y \succ y_b|q)$ across all prompts to determine the final win rate for ranking all evaluated LLMs. We directly use the model's output from the AlpacaEval official repo.[3] We also use our ptoposed method to calibrate AE1. Since AE1 uses GPT4 to compare two responses pairwisely and uses its output logits to compute

---

[3]https://github.com/tatsu-lab/alpaca_eval/tree/main/results

the preference probability $p(y \succ y_b|x)$. Thus, following Eq. 1, we employ the inverse function of sigmoid to calculate the reward margin $r - r_b$. We then utilise the LWR to directly predict the biased reward margin given the length margin $|y| - |y_b|$.

**Hyper-parameters**     Two key hyper-parameters for our proposed method, `RC-Mean` and `RC-LWR`, are the threshold $d$ and the bandwidth $f$. (1) **Determine the threshold** $d$: The way we use to determine the threshold distance $d$ in Eq. 6 is as follows: (1) Taking the length bias calibration as an example, given a dataset of data pairs to calibrate $\{(x_1^i, x_2^i, r_\theta(x_1^i), r_\theta(x_2^i)\}$, we first calculate the average of $||x_1^i| - |x_2^i||$, noted as $\text{AVG}(||x_1| - |x_2||)$, (2) then we set $d = \text{AVG}(||x_1| - |x_2||)/4$, which is set by intuition. And if the number of data points within the neighborhood is smaller than 10, we do not calibrate that pair as the results may be overly affected by specific data points. This set of hyper-parameters is employed for all three different experimental settings. (2) **Determine the bandwidth** $f$ **for LOWESS**: this hyper-parameter is closely related to the assumption. Regarding the experiment of RM as LLM evaluators and RM for LLM alignment, we consider the sufficient density assumption valid since there are 150k samples and 300k samples, respectively. We thus set the bandwidth as default as in `statsmodels.api`, $f = \frac{1}{3}$, and the number of iterations is 3. (Though we use all scored data points for LOWESS regression, we show in Fig. 12 and Fig. 13 that the methods yield improvements with only hundreds of data points.) Regarding the experiment on the RewardBench, there are only 2,985 data pairs, resulting in 5970 data points. Therefore, the sufficient density assumption may be invalid for some length intervals, so we choose a larger bandwidth $f = 0.9$. Importantly, all hyper-parameters relevant to the reward calibration haven't been tuned for specific RM. For RewardBench experiments, hyper-parameters for all **33** BT-based RMs and **46** DPO-based reward models are the same, and for AlpacaEval calibration, the hyper-parameters used for 8 models are the same. We argue that this further validates the generality of our proposed methods. In practice, one could conduct a more comprehensive hyper-parameter search for a specific RM.

**Computational Environments**     All experiments about reward calibration are performed with a single CPU. Experiments for getting the reward model's score on AlpacaEval's output are conducted using a NVIDIA H100 GPU. Experiments regarding using DPO for LLM alignment use 8 NVIDIA H100 GPU and costs about 2-3 hours for the DPO training.

## D    EXTRA EXPERIMENTAL RESULTS

**Calibration Results For DPO-based Reward Models**     In addition to calibrating the traditional BT-based RMs, we calibrate length bias with DPO-based reward models. Formally, given a pair of completions $(y_1, y_2)$ to the prompt $q$, the preference of the DPO-trained model $\pi_\theta$ could be determined by comparing the following log-ratios: $\frac{\pi_\theta(y_1|p)}{\pi_{ref}(y_1|p)}$ and $\frac{\pi_\theta(y_2|p)}{\pi_{ref}(y_2|p)}$, where $\pi_{ref}$ is the reference model during the DPO training. We directly employ the official rewarding results from the RewardBench leaderboard. The reasons why we choose to present the results for DPO-based models in the appendix instead of the main text are twofold: (1) The rewarding results for DPO-based model are more noisy as some of their reference models are not available; (2) To the best of our knowledge, DPO-tuned models are more frequently used as chat models or instruct models, and are rarely employed to indicate the preference with the implicit reward. The results are shown in Fig. 6. Our observations are as follows: (1) The overall reward performance of DPO-based models is inferior to that of BT-based reward models, as well as their improvements after calibration. (2) The proposed methods, `RC-LWR` and `RC-LWR`, are effective in most cases, achieving average performance gains of 1.60 and 1.68, respectively. (3) The method `RC-Mean` can lead to improvements for most models. However, it can also result in performance collapse. For instance, after calibration, the performance of `Qwen1.5-72B-Chat` decreases from 71.5 to 49.9. (4) The `Length Penalty` shows no calibration effect for most DPO-based models because the absolute values of log ratios, used as implicit rewards, are usually high. This finding further emphasizes that the penalty weight requires more comprehensive research in practical applications and is highly dependent on the specific reward scales of the model and the length distribution of the dataset.

**Robustness to the choice of bandwidth**     We investigate the extent to which the calibration effect is influenced by different choices of bandwidth $f$. The experiments are conducted using RewardBench

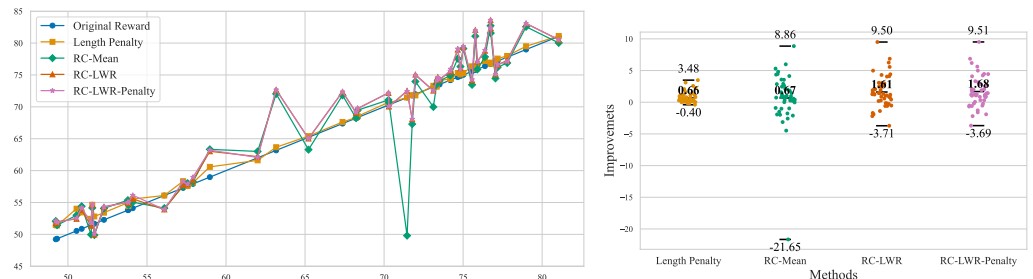

Figure 6: Results of different calibration algorithms on the RewardBench benchmark for DPO-based reward models are shown in two charts. **Left**: the line chart demonstrates the RewardBench score before calibration (x-axis) and after calibration (y-axis) of different algorithms for different models. **Right**: the scatter plot highlights the performance gains achieved by different calibration methods, annotated with the maximum, average, and minimum values.

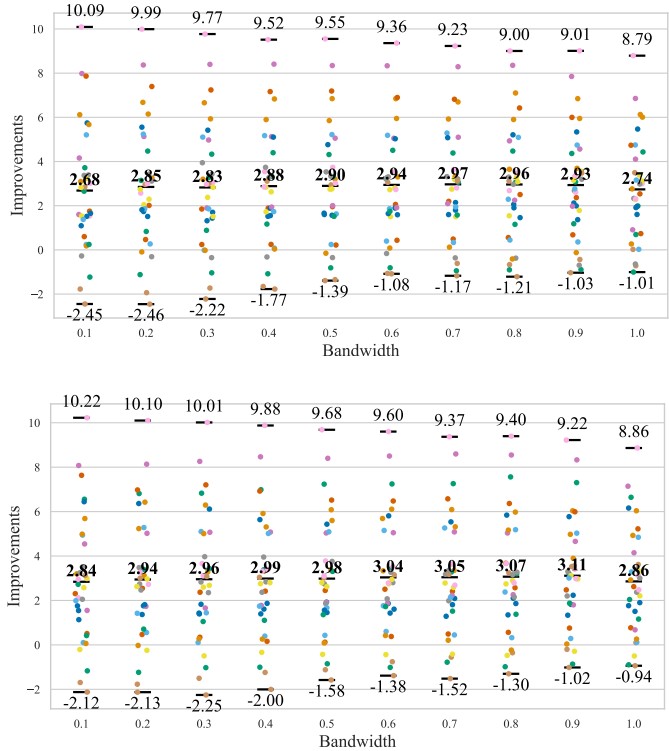

Figure 7: Calibration results, shown in the improvements, of `RC-LWR` (**top**) and `RC-LWR-Penalty` (**bottom**) with different bandwidth $f$ on the RewardBench dataset.

because, for the RewardBench setting, there are only 6k data points, and the smaller number of total data points for calibration may increase sensitivity to variations in bandwidth. The overall results for different bandwidths for both `RC-LWR` and `RC-LWR-Penalty` are presented in Fig. 7. Our empirical findings are as follows: (1) Overall, the proposed methods demonstrate robustness across varying bandwidths, achieving similar average performance gains when the bandwidth ranges from 0.1 to 1.0. For `RC-LWR` (top), the average gain ranges from 2.68 to 2.96, while for `RC-LWR-Penalty`, it ranges from 2.84 to 3.11. A slight performance improvement is observed when $f$ increases from 0.1 to 0.9. (2) Although the average performance gains remain relatively consistent, the maximum improvements and degradations exhibit more notable trends. Specifically, as the bandwidth increases, both the maximum improvements and degradations diminish. These results suggest that calibration

can be controlled through bandwidth selection, with larger bandwidths yielding more stable calibration effects.

**Controlling calibration effects**   One critical assumption for our proposed method is that the underlying gold reward is independent of the biased characteristic. We also showcase that our method could reduce the correlation between rewards and the characteristic of interest to 0. However, the independence assumption may be invalid if one focuses on a specific subset of instructions. For example, the independence assumption is considered valid given a general prompt set. But if we add "generate contents as concise as you can" at the end of every prompt, thus the underlying true reward should be positively correlated with the output length for this prompt set. Therefore, a more practical calibration method should be controllable for users in different application scenarios. We show that our proposed method can also be controlled by introducing a hyper-parameter named calibration constant $\gamma$ to the Eq. 7:

$$\hat{\Delta}^*_{r_\theta}(x_1, x_2) = \Delta_{r_\theta}(x_1, x_2) - \gamma \times (\hat{r}_\theta(c(x_1)) - \hat{r}_\theta(c(x_2))) \qquad (8)$$

We validate the effects of $\gamma$ using two experimental settings: RM on the RewardBench and RM for LLM alignment. The ablation results on the RewardBench dataset are presented in Fig.8 , while those for the LLM alignment setting are shown in Table 4 . A noteworthy statistic for the RewardBench dataset is that simply selecting the shorter response yields a 60% accuracy. Thus, the ideal correlation between rewards and response length should be slightly negative. The results in Fig. 8 support this notion: if $\gamma = 1$ results in zero Spearman correlation as indicated in the Table. 3, the optimal $\gamma$ should exceed 1. Our findings indicate that the best average performance gain occurs when $\gamma = 1.4$. More importantly, the results across both settings demonstrate that the calibration effect can be smoothly controlled through the calibration constant. This significantly enhances the practical utility of the proposed method, as one can adjust this constant to achieve superior rewarding performance if certain prior knowledge about the correlation between rewards and the characteristic of interest is known.

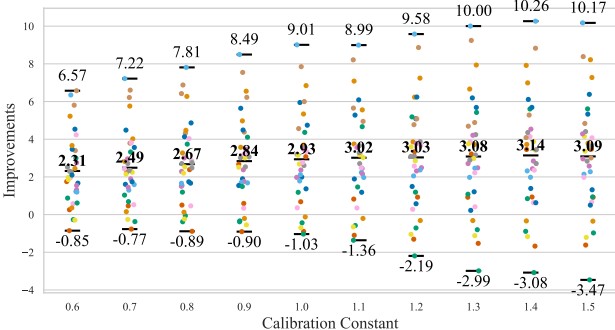

Figure 8: Ablation results of different calibration constant on the RewardBench for BT-based RMs.

**Length distribution**   Defining the length margin as the absolute difference in length between two completions, the cumulative distribution function of the length margin on four different datasets is shown in Fig. 9. As shown in the Fig. 9, different datasets demonstrate distinct length margin distributions. For Ultrafeedback-Llama3 and Ultrafeedback-Gemma2, about 80% of data pairs have a length margin that is less than 600, which is reasonable because the compared completions are sampled from the same LLM. And the RewardBench dataset demonstrates a similar pattern. In contrast, the length margin on the AlpacaEval benchmark is a lot longer, with only about 40% of the data points having a length margin that is less than 600. Notably, the simple `Length Penalty` method works well in calibrated RM on RewardBench and calibrated RM for LLM alignment settings. This may be because the length margin in those two settings is smaller, thus leading to moderate penalization. However, when length margins are more significant, more meticulous adjustments for penalty weight are required as they are more sensitive. In summary, the baseline `Length Penalty` is already strong when RMs exhibit strong length bias. However, both reward scales and length margins should be considered to adjust the penalty weight, impairing its practicability. In contrast, the proposed method `RC-LWR` is robust to different hyper-parameter choices, yielding stable improvements in different settings.

Table 4: Results of using Length-Calibrated rewards for LLMs' alignment with different calibrate constant $\gamma$ in `RC-LWR`. We apply these methods with four different LLM–RM configurations. The top reports the Length-controlled win rate, and the bottom showcases the average length of generations on AlpacaEval dataset.

| Algorithm | 0 | 0.2 | 0.4 | 0.6 | 0.8 | 1.0 |
|---|---|---|---|---|---|---|
| **Length-Controlled Win Rate** | | | | | | |
| `Llama-3-8B-Fsfaix` | 41.82 | 42.59 | 46.39 | 48.12 | 50.25 | **51.37** |
| `Llama-3-8B-GRM` | 41.00 | 43.54 | 46.44 | 49.76 | 49.61 | **50.49** |
| `gemma2-9b-Fsfaix` | 62.79 | 63.96 | 62.69 | 67.57 | 68.87 | **69.54** |
| `gemma2-9b-Grm` | 63.21 | 67.23 | 66.95 | 67.27 | 68.27 | **70.45** |
| **Average Character Length on Alpaca Benchmark** | | | | | | |
| `Llama-3-8B-Fsfaix` | **2404** | 2365 | 2340 | 2194 | 2028 | 1867 |
| `Llama-3-8B-GRM` | **2438** | 2375 | 2303 | 2194 | 1986 | 1858 |
| `gemma2-9b-Fsfaix` | 2365 | 2321 | **2570** | 2028 | 1725 | 1798 |
| `gemma2-9b-Grm` | **2106** | 2117 | 1939 | 1860 | 1638 | 1780 |

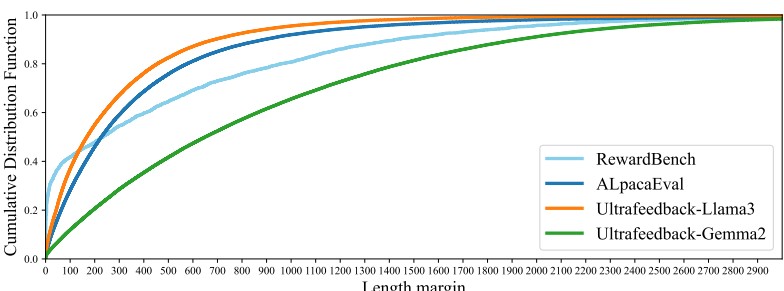

Figure 9: The cumulative distribution function of the length on different datasets used in our experiments.

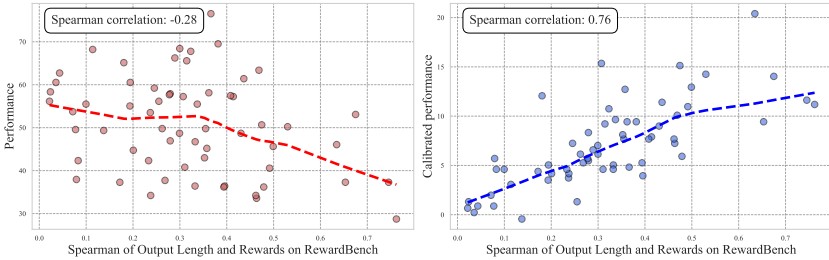

Figure 10: Based on the Chat Hard subset of RewardBench, we plot how the performance (left) or improvement brought (right) by `RC-LWR` vary with the RM's length preference, which is represented by the Spearman score between output length and rewards on RewardBench.

**Reward models with more substantial bias are improved more.** Results in Fig. 4 motivate us to investigate how calibration gains vary for RMs with different length bias. Therefore, we selected Chat Hard, the most challenging subset of RewardBench, for our investigation. Results are shown in Fig. 10. We use the Spearman correlation to represent the RMs' length preference on the x-axis; we then plot the RM's performance and `RC-LWR` improvements, different points in Fig. 10 refers to different RMs. In Fig. 10 left, we observe that the RM's performance on Char Hard negatively correlates with the RM's length preference, achieving $-0.28$ correlation. (Note that a rule-based RM that always chooses the shorted sentence could achieve an accuracy of 90% on Chat Hard subset.) In Fig. 10 right, we observe a strong positive correlation between the RM's length preference and its improvement on the Chat Hard subset, achieving $0.76$ correlation score. Note that similar trends are found between length preference and overall performance improvement of RewardBench with $0.47$ Spearman Correlation score, both indicating that Reward Models with stronger length preference are improved more by the proposed `RC-LWR` method.

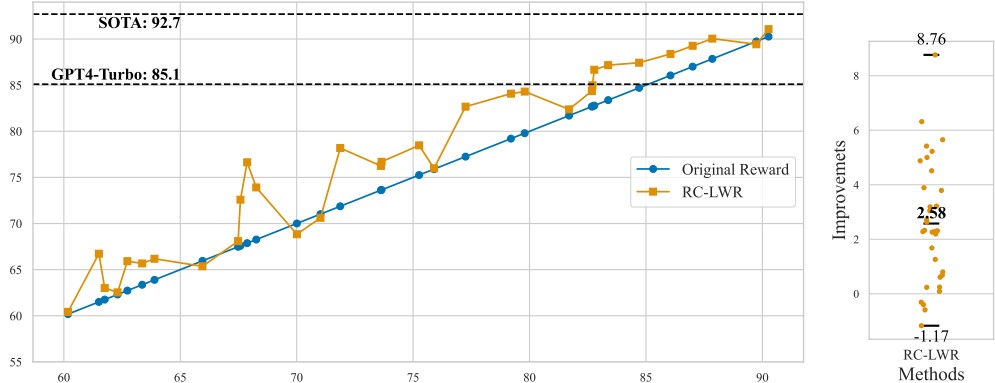

Figure 11: Results of calibrating with output length and markdown features simultaneously.

**Calibrate for more than one characteristic**  One straightforward extension to calibrate with more than one characteristic is to employ multi-dimensional Locally Weighted Regression. For example, if we consider calibrating two characteristics $c_1$ and $c_2$ simultaneously, the dataset employed to do the LOWESS regression could be described as $D = \{(c_1(x_j), c_2(x_j), r_\theta(x_j))\}_{j=1}^N$. Then, the Euclidean distance is employed to measure the distance between different data points. Based on this, we extend our method to directly calibrate rewards with output length and markdown features. Because these two characteristics have very different scales, we first normalize their values by Z-score normalization. The calibration results are shown in the Fig. 11. We observe that: (1) the proposed method `RC-LWR` could directly generalize to calibration for multiple characteristics, yielding a 2.58 average performance gain, (2) the gain is superior to solely calibrating markdown bias (1.86) but inferior to solely calibrating length bias (2.93). Possible reasons for no synergy effect include: a. The two characteristics of interest here, length and markdown features, may correlate, and the method does not touch on thiat; b. the Euclidean distance used here considers two characteristics equivalent.

**Data efficiency**  LOWESS is a non-parametric method that requires ground truth labels to compute residuals during its robustifying iterations, making it logical to utilize all available data points for calibration. We demonstrate that even with a moderate dataset size (2,985 preference pairs in RewardBench), calibration yields significant improvements. To explore this further, we conduct a stress test on the number of data points used for calibration. Focusing on the RewardBench dataset, our experimental procedure involves sampling a subset of the dataset, performing calibration on the selected subset, and using accuracy as the evaluation metric. It is important to note that accuracy is not directly comparable to the RewardBench score in our main experiments. The RewardBench score is the average accuracy across four subsets (Chat, Chat Hard, Safety, and Reasoning), which vary in size. We define the subset size as a fraction of the original dataset, ranging from 0.01, 0.05, 0.1, 0.2, 0.3, 0.4, to 0.5. For each subset size, we run ten experiments with different random seeds and average the performance across these seeds. The results for different subset size for `RC-LWR-Penalty`, `RC-LWR`, `RC-Mean` are shown in the Fig. 12, Fig. 13 and Fig. 14, respectively. We observe that the intuitive method, `RC-Mean`, achieves increasing accuracy as the subset size grows, while `RC-LWR` and `RC-LWR-Penalty` exhibit more stable improvements across different sizes, demonstrating that our method is not heavily constrained by data density.

**Further analysis about the chosen length penalty baseline**  In our main experiment results, we follow previous research works to set $\alpha = 0.001$, which may raise concerns that the $\alpha$ is not optimal for the selected tasks and reward models. Therefore, we first do ablation studies with different $\alpha$ values from $\{0.1, 0.05, 0.02, 0.01, 0.005, 0.002, 0.001, 0.0005, 0.0002, 0.0001\}$. To further decouple the effect of the fact that different reward models have different reward distributions, we propose to first z-normalize the reward and then apply the `Length-Penalty` method. The results of these two methods are demonstrated in Fig. 15. According to Fig. 15, the chosen value $\alpha = 0.001$ achieves the best performance compared with other values (Original line). The Z-normalization trick could slightly boost the performance to `2.75`, which is reasonable because it alleviates the issue that different reward models have different distributions. However, both of them underperform the proposed

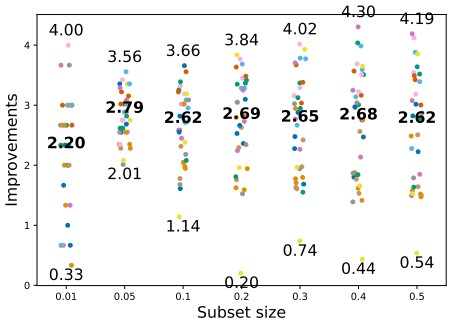

Figure 12: Calibration results with different subset size for `RC-LWR-Penalty` method

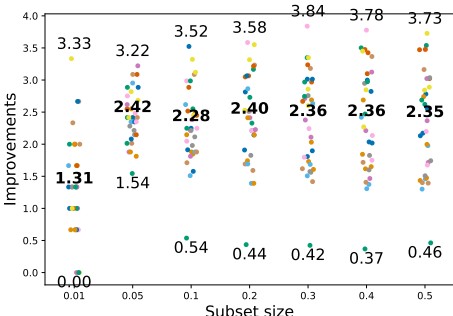

Figure 13: Calibration results with different subset size for `RC-LWR` method

method. Furthermore, we argue that the optimal $\alpha$ is challenging to search for the following reasons: (1) both the reward distribution of the model and the length distribution of the task affect $\alpha$, asking users to search different $\alpha$ per reward model and tasks repeatedly; (2) the method is very sensitive to the value of $\alpha$, for some values the model may even collapse. On the contrary, our proposed method could robustly adapt to different reward models and tasks.

**Reward calibration v.s. Reward Ensemble**  As discussed in our related work section, previous works can utilise ensemble techniques to mitigate the potential bias. We thus try to compare our proposed `RC-LWR` with the ensemble. Specifically, we use normalize-then-average to aggregate rewards from different reward models. We first select the top nine models from the RewardBench Leaderboard as of Aug and then aggregate them. The results are shown in the Fig. 16. In Fig. 16, the number in the cell $[i, j]$ denotes the ensemble performance of model $i$ and model $j$ where $i$ and $j$ are the model's rank on the RewardBench leaderboard. The last row is the calibrated performance of models. We find that the calibration outperforms most ensemble results except for ensembles with a much stronger model. And intuitively, ensemble-based methods may not address common bias across models. For example, if these models overly prefer lengthy outputs, their ensemble model will also prefer lengthy output.

**Visualization**  To better illustrate how our proposed method shifts the reward distribution of the model and removes the length bias, we present the kernel density plot (KDE) of rewards (y-axis) w.r.t length (x-axis) for different reward models, calibration methods and hyper-parameters. Specifically, we select ten different reward models (Fig. 17– 26). For each reward mode, we visualize four distributions based on RewardBench dataset: (a) The first (left) figure demonstrates the original distribution between the reward and character length of the example; (b) The second figure demonstrates the calibrated distribution of our proposed `RC-LWR` method; (c) The third figure demonstrates the calibrated distribution of the proposed variant of `RC-LWR` (Eq.  8) with $\gamma = 1.25$; (d) the last (right) figure illustrates the calibrated distribution of the `Length-Penalty` baseline with $\alpha = 0.001$. In each figure, the colour represents the density of data points. The deeper the colour,

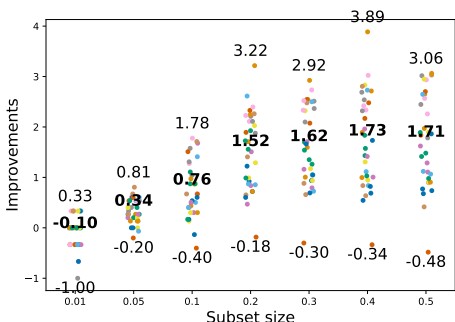

Figure 14: Calibration results with different subset size for `RC-Mean`

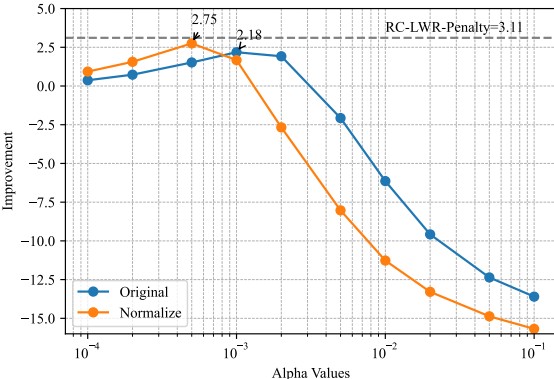

Figure 15: The performance of `Length-Penalty` baseline with different $\alpha$ values in the RewardbBnech setting. Two different variants are considered here. Original means that we directly add the penalty term to the Reward Model. Normalize means we first z-normalise the rewards and then apply the `Length-Penalty` baseline.

|            | 1     | 2     | 3     | 4     | 5     | 6     | 7     | 8     | 9     |
|------------|-------|-------|-------|-------|-------|-------|-------|-------|-------|
| 1          | 90.26 | 92.25 | 90.09 | 89.66 | 89.56 | 89.56 | 89.66 | 88.93 | 88.31 |
| 2          | 92.25 | 89.74 | 91.43 | 89.56 | 89.08 | 89.29 | 89.57 | 88.71 | 89.03 |
| 3          | 90.09 | 91.43 | 87.85 | 89.00 | 88.65 | 88.70 | 87.68 | 87.16 | 87.22 |
| 4          | 89.66 | 89.56 | 89.00 | 87.00 | 86.77 | 86.72 | 86.17 | 86.59 | 86.14 |
| 5          | 89.56 | 89.08 | 88.65 | 86.77 | 86.05 | 86.23 | 86.48 | 85.86 | 85.74 |
| 6          | 89.56 | 89.29 | 88.70 | 86.72 | 86.23 | 84.71 | 85.70 | 85.60 | 84.82 |
| 7          | 89.66 | 89.27 | 87.68 | 86.17 | 86.48 | 85.70 | 83.38 | 85.36 | 84.49 |
| 8          | 88.93 | 88.71 | 87.16 | 86.59 | 85.86 | 85.60 | 85.36 | 82.78 | 84.48 |
| 9          | 88.31 | 89.03 | 87.22 | 86.14 | 85.74 | 84.82 | 84.49 | 84.48 | 82.70 |
| Calibrated | 92.08 | 89.53 | 91.75 | 90.19 | 88.53 | 88.12 | 88.04 | 86.34 | 85.67 |

Figure 16: Ensemble performance and calibrated performance

the denser the data points. The red dashed line illustrates the LOWESS smoothing for the KDE plot. According to these figures, we find that: (1) As discussed in our introduction, the length bias exists for nearly all selected ten reward models. (Note that the ideal correlation on RewardBench should be slightly negative.) And the correlation is non-linear. (2) The red dashed lines become almost horizontal after `RC-LWR` calibration, indicating that our method removes the reward's correlation with length smoothly and uniformly. (3) By setting $\gamma = 1.5$, we can push the correlation of rewards and length to be negative, showcasing the potential of our method to inject desirable bias into reward models. (4) Regarding the `Length Penalty` baseline, though the Spearman correlation between reward and length decreases, the length bias is not removed. For models with no length bias (e.g., `NCSOFT/Llama-3-OffsetBias-RM-8B`), the length penalty method will push the reward to prefer shorter responses.

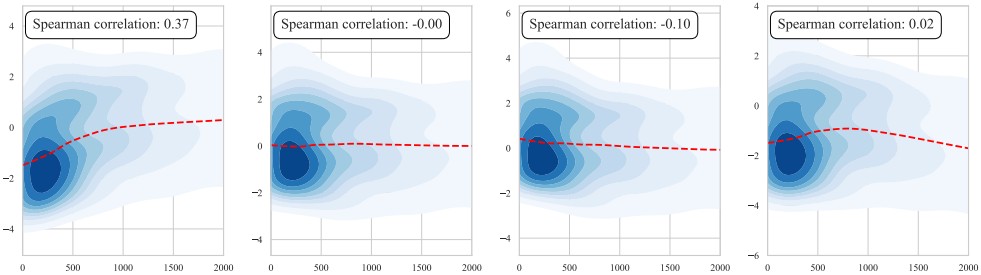

Figure 17: `internlm/internlm2-20b-reward`

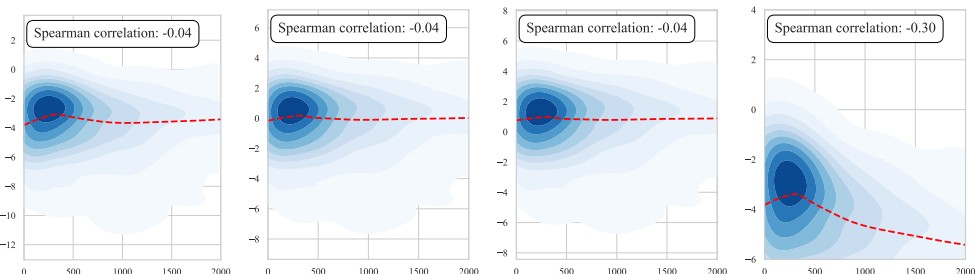

Figure 18: `NCSOFT/Llama-3-OffsetBias-RM-8B`

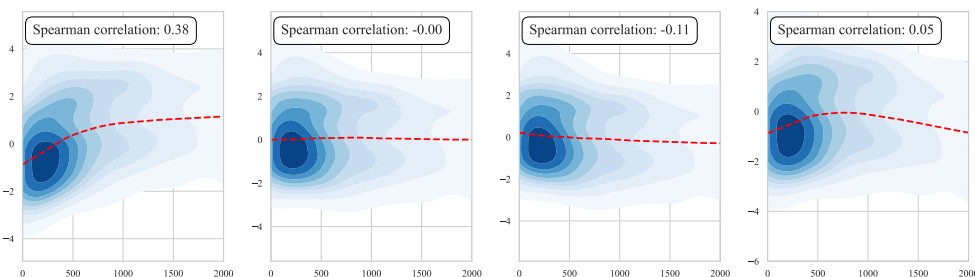

Figure 19: `internlm/internlm2-7b-reward`

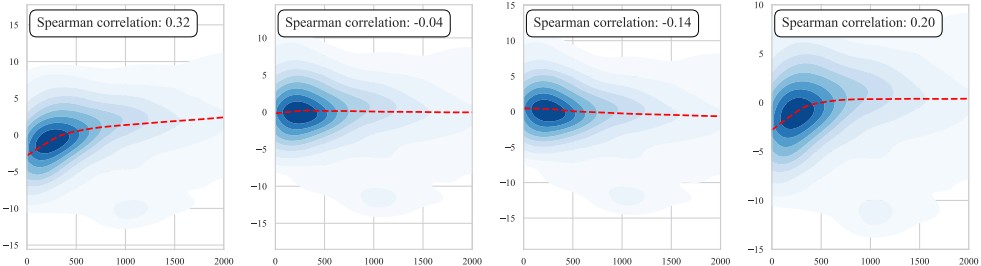

Figure 20: `Ray2333/GRM-llama3-8B-sftreg`

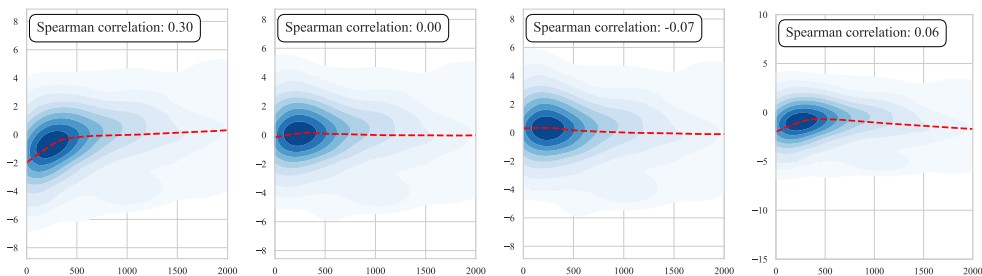

Figure 21: `Ray2333/GRM-llama3-8B-distill`

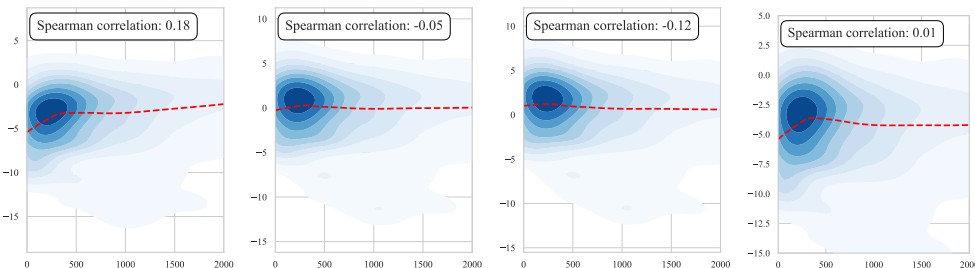

Figure 22: `sfairXC/FsfairX-LLaMA3-RM-v0.1`

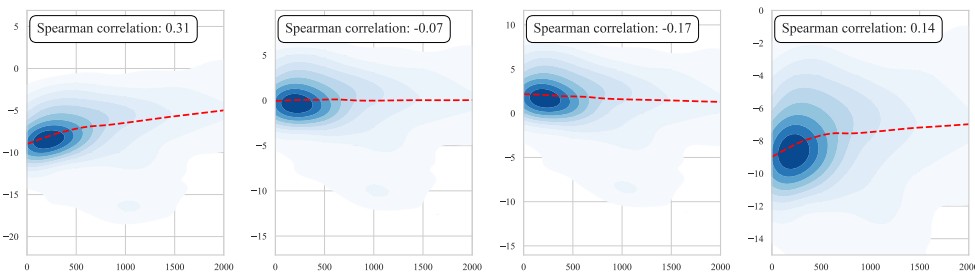

Figure 23: `CIR-AMS/BTRM_Qwen2_7b_0613`

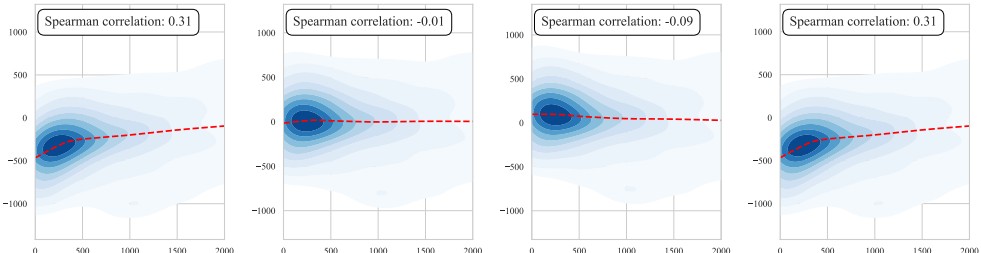

Figure 24: `openbmb/Eurus-RM-7b`

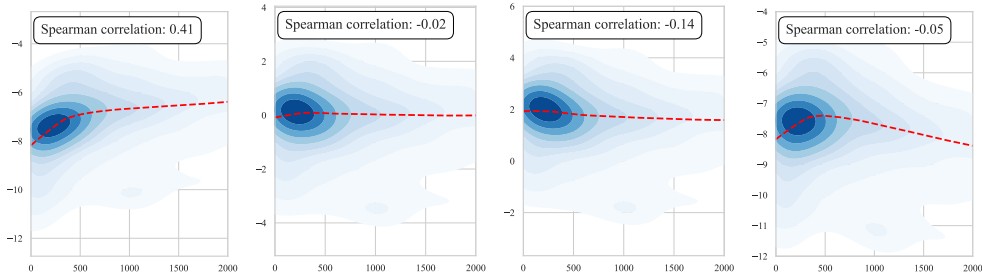

Figure 25: `Nexusflow/Starling-RM-34B`

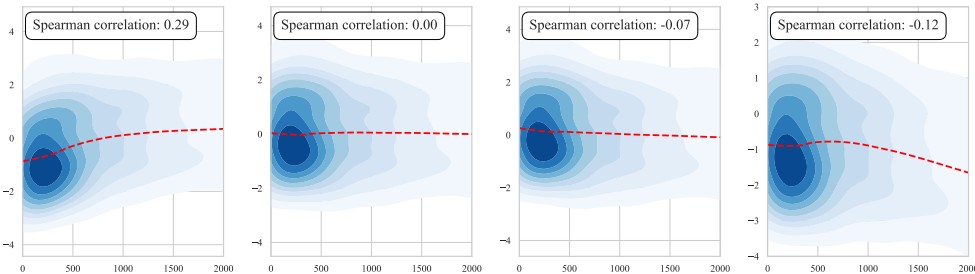

Figure 26: `internlm/internlm2-1_8b-reward`

