# OpenReview forum: "Post-hoc Reward Calibration: A Case Study on Length Bias"
_ICLR.cc/2025/Conference — ICLR 2025 Poster_

### Official Review · Reviewer_aRiJ · 2024-10-28

**Soundness:** 4
**Presentation:** 4
**Contribution:** 3
**Rating:** 8
**Confidence:** 3

**Summary:**

This paper proposes a calibration method for correcting bias (in particular length bias) in already trained LLM reward models (RM) on datasets. Assuming that the reward can be decomposed into an aligned reward and a bias reward term, the proposed method estimates the bias reward term using locally weighted regression (LWR) or a more robust variant (LOWESS). Then the correct reward is obtained by simply subtracting the estimated bias term. Experiments in RMBench, RM as Judge, and RM alignment show that the proposed method (RC-LWR) is able to effectively and robustly remove length bias compared with the common approach of adding a length penalty during RM training. Adding a length penalty requires a task-dependent penalty weight, whereas RC-LWR robustly adapts to different tasks and datasets.

**Strengths:**

- The proposed method, RC-LWR, is post-hoc and so does not require intervention during the training of the RM. It is also fast to compute. This means you can make many post-hoc corrections and sweep over hyper parameters all without needing to retrain. This is a huge advantage over adding a length penalty or other penalty term during RM training. This means you can try to correct for as many different biases as you want to do and try out combinations of them all post-hoc.

- RC-LWR is also robust to the curvature of the bias term due to using locally weighted regression, and only relies on the bias being locally close to linear.

- The paper is very clear and easy to follow.

**Weaknesses:**

- A slight weakness is the majority focus on length bias. The experiments pretty much all focus on length bias, with one generalization task to markdown features.

- Another slight weakness is that only the combination of length penalty + RC-LWR was tested. A more interesting direction would be to test a combination of two biases, and whether you can apply the method multiple times to remove multiple biases.

**Questions:**

Typo between lines 224-225 “Based one LWR…”

---

> ### Author Response · Authors · 2024-11-18
> **Responses to reviewer aRiJ**
>
> Thanks for recommending our paper! Your comments are highly aligned with our paper. We have adjusted our paper in this revision according to your suggestions. We hope the following comments can answer your questions.
>
> >  A slight weakness is the majority focus on length bias. The experiments pretty much all focus on length bias, with one generalisation task to markdown features.
>
> Thanks for the comments. We would still like to highlight that length bias is one of the most prevalent biases for LLM evaluation, RM training, and RLHF. The phenomenon arises from the confluence of multiple contributing factors in the output. For example, the model's use of listing, markdown style, URL, or complex tokens can all correlate to length bias.
>
> Compared with existing works focusing on length bias, our method can also effectively address the length bias issue without extra training or data and be readily applicable to a wide range of scenarios. We extend the proposed method to the md feature to demonstrate its generality. And robustly extending it to other features is one critical part of our future work\!
>
> >  Another slight weakness is that only the combination of length penalty \+ RC-LWR was tested. A more interesting direction would be to test a combination of two biases and determine whether you can apply the method multiple times to remove multiple biases.
>
> Thanks for your suggestion\! We provided the relevant results in our submission (Fig. 11). Specifically, we extend the proposed method to calibrate more than one characteristic by employing multidimensional Locally Weighted Regression, with every dimension representing different biases. However, our results show that simultaneously calibrating two features (length and md features) does not lead to stronger improvements. One primary reason could be that different biases are not independent and may correlate (The same concern also holds when removing different biases sequentially). And modelling their correlation is non-straightforward.
>
> Overall, we agree with your idea to extend the current method to calibrate other/multiple biases. We think the current version, working well for length bias, already contributes well to the LLM community and we would love to explore your proposed directions in our future work\!

---

> > ### Comment · Reviewer_aRiJ · 2024-11-22
> > **Thank you for your responses**
> >
> > Thank you for your response. I was already in favour so I maintain my accept score.

---

### Official Review · Reviewer_fLWP · 2024-10-29

**Soundness:** 2
**Presentation:** 3
**Contribution:** 2
**Rating:** 6
**Confidence:** 4

**Summary:**

This paper focuses on the reward modeling process in Reinforcement Learning from Human Feedback (RLHF). The authors highlight the issues of bias and spurious correlations in the training data and propose a method called post-hoc reward calibration to address these limitations. Specifically, they assume the reward model can be decomposed into two components and use the evaluation set to estimate the bias term through locally weighted regression. The experimental results demonstrate the method's effectiveness.

**Strengths:**

1.	The authors effectively identify common limitations in reward models and propose a method to enhance their reliability.

2.	The proposed method is claimed to be computationally efficient.

3.	Experimental results confirm the method's effectiveness.

**Weaknesses:**

1.	Assumptions: The proposed method depends heavily on several assumptions, such as (1) decomposition of the biased reward, (2) independence of the bias characteristic, (3) sufficient density, and (4) Lipschitz continuity. These assumptions may not hold in all scenarios.

2.	Difficulty in Determining $d$: In some scenarios, determining dd is difficult. While LWR can estimate Equation 5, linear approximation may often be inaccurate, necessitating further analysis.

3.	Simple Baselines: The baselines used are relatively simple. Comparisons with more complex baselines, such as reward model engineering in reference, are needed.

4.	Demonstrations Needed: Additional examples should be provided to showcase the method's effectiveness in various situations.

6.	Bias Exploration: It would be beneficial to explore other biases beyond 'length' and 'style', given that significant improvements are mostly seen in 'length bias'.

**Questions:**

1.	Bias and Spurious Correlation: How does the characteristic bias $c$ lead to spurious correlations? Examples would facilitate a better understanding for the reader.


2.	Hyperparameter Selection: What guidelines are there for selecting hyperparameters in different scenarios, such as $\beta_0$ and $\beta_1$?

3.	Other Types of Bias: Are there experiments demonstrating bias beyond length and style?

4.	Fig. 5 x-axis: What does the x-axis represent in Figure 5? Providing clarity would help readers interpret the results more effectively.

5.	There are several methods that can mitigate length bias, particularly in DPO. Have you considered simple baselines such as simPO?

---

> ### Author Response · Authors · 2024-11-18
> **Responses to reviewer fLWP Part 1 /3**
>
> Thanks for your comments and your questions\! Please see our pointwise responses to see if we could address your concerns.
>
> > Assumptions: The proposed method depends heavily on several assumptions, such as (1) decomposition of the biased reward, (2) independence of the bias characteristic, (3) sufficient density, and (4) Lipschitz continuity. These assumptions may not hold in all scenarios.
>
> Yes, we agree that these assumptions may not hold in all scenarios. However, we have verified the effectiveness of our method across various reward models and scenarios **where these assumptions do not perfectly hold**. For example, in RewardBench, simply selecting the shorter responses would yield 60% accuracy; thus, the ideal correlation between reward and length should be slightly negative. On the contrary, in the RM-as-LLM-evaluator setting, the ideal correlation should be positive, according to the ChatbotArena.
>
> Furthermore, these assumptions are not that difficult to deal with/satisfy in practice.
> 1. For the independence assumption, we can introduce $\gamma$ to control the calibration effect (please see the common concern in General Comment and line 1029 - 1049 in our paper).
> 2. Secondly, the sufficient density assumption is proposed because we may require sufficient data points for estimation: in the appendix, we perform a stress test to measure how much data we need for estimation (Fig. 12-14) and demonstrate that our proposed method can provide stable calibration even with only a few hundreds of data points.
> 3. Thirdly, the Lipschitz continuity just assumes that the reward value is a slow-varying function of $c$, i.e., it will not change dramatically with $c$. For example, for calibrating length bias, the function is not even continuous because we define length as the character length of the output. Still, the introduced method is effective regarding length bias.
>
> We will add more discussions to clarify the essence of these assumptions.
>
> > Difficulty in determining d: In some scenarios, determining dd is difficult. While LWR can estimate Equation 5, linear approximation may often be inaccurate, necessitating further analysis.
>
> Yes, determining d in equation 6 is challenging in practice, as explained in our submission (lines 192-198). Therefore, we introduce LWR to estimate the equation 5\. For LWR, the only hyper-parameter is the bandwidth $f$ that determines the portion of the dataset used for locally weighted regression.  Working with a local neighbourhood, LWR is broadly considered a good smoothing method for functions with unknown shapes.  Moreover, our analysis in the appendix reveals that the proposed method is robust to different bandwidths $f$ of LWR.
>
> > Demonstrations Needed: Additional examples should be provided to showcase the method's effectiveness in various situations.
>
> First, we want to emphasise that the effectiveness of our proposed method is demonstrated across three different settings, covering dozens of reward models (RewardBench setting), hundreds of LLMs (LLM evaluator setting) , and various LLM-RM combinations.
>
> Then, we provide extra figures illustrating how the distribution of reward w.r.t. length changes before and after calibration to offer more insight into the reward calibration's result. The supplementary figures are in the appendix of our revised paper (Fig. 17--26).

---

> ### Author Response · Authors · 2024-11-18
> **Responses to reviewer fLWP Part 2 / 3**
>
> > Simple baselines: The baselines used are relatively simple. Comparisons with more complex baselines, such as reward model engineering in reference, are needed.
>
> To the best of our knowledge, the length penalty is the only baseline comparable to our method (i.e., it does not require extra data, extra models, or extra training and is applicable in various scenarios).  Besides the length penalty baseline, in RMs as LLMs evaluator setting, we also compare with length-controlled regression (LC) proposed by Dubois et al. (2024) and was utilized to produce the AlpacaEval 2.0 leaderboard. LC is specifically designed for the AlpacaEval leaderboard and thus needs instruction_id and model_id for regression. Our method achieves comparable calibration performance without such information with LC on RM and GPT4 judges.
>
> Because in our related work section, we mainly mention relevant works that ensemble reward models to mitigate bias, we supplement a baseline that ensemble rewards from different reward models. Specifically, we select the Top 9 models from the leaderboard as of Aug. and enumerate all possible ensembles between the two models. Their ensemble results are shown in the following Table. The cell\[i,j\] denotes the performance by ensembling the model_i and model_j, where i and j are their ranks on the RewardBench leaderboard.
>
>  The final row shows the performance after our method's calibration.
>
> The calibration **surpasses most ensemble results** except for ensembles with a much more robust model (The ensemble results that outperform the calibration method are bold, primarily the results that ensemble with the top-1 model).
> Intuitively, ensemble-based methods may not address common biases across models. For example, if these models overly prefer lengthy outputs, their ensemble model will also prefer lengthy outputs.
>
> | Model i + Model j | 1         | 2         | 3     | 4     | 5         | 6         | 7         | 8         | 9         |
> | ----------------- | --------- | --------- | ----- | ----- | --------- | --------- | --------- | --------- | --------- |
> | 1                 | 90.26     | **92.25** | 90.09 | 89.66 | **89.56** | **89.56** | **89.66** | **88.93** | **88.31** |
> | 2                 | **92.25** | **89.74** | 91.43 | 89.56 | **89.08** | **89.29** | **89.57** | **88.71** | **89.03** |
> | 3                 | 90.09     | **91.43** | 87.85 | 89.0  | **88.65** | **88.7** | 87.68     | **87.16** | **87.22** |
> | 4                 | 89.66     | **89.56** | 89.0  | 87.0  | 86.77     | 86.72     | 86.17     | **86.59** | **86.14** |
> | 5                 | 89.56     | 89.08     | 88.65 | 86.77 | 86.05     | 86.23     | 86.48     | 85.86     | **85.74** |
> | 6                 | 89.56     | 89.29     | 88.7  | 86.72 | 86.23     | 84.71     | 85.7      | 85.6      | 84.82     |
> | 7                 | 89.66     | 89.27     | 87.68 | 86.17 | 86.48     | 85.7      | 83.38     | 85.36     | 84.49     |
> | 8                 | 88.93     | 88.71     | 87.16 | 86.59 | 85.86     | 85.6      | 85.36     | 82.78     | 84.48     |
> | 9                 | 88.31     | 89.03     | 87.22 | 86.14 | 85.74     | 84.82     | 84.49     | 84.48     | 82.7      |
> | Calibrated Perf   | 92.08     | 89.53     | 91.75 | 90.19 | 88.53     | 88.12     | 88.04     | 86.34     | 85.67     |
>
> We have also supplemented this in our revised paper (lines 1222 - 1231).
>
> > Bias Exploration: It would be beneficial to explore other biases beyond 'length' and 'style', given that significant improvements are mostly seen in 'length bias'.
> >
> > Other Types of Bias: Are there experiments demonstrating bias beyond length and style?
>
> So far, we have mainly explored the two widespread biases of RMs: the length and style of the output. We respectfully believe that addressing length bias in various scenarios is already an essential contribution to the LLM community since length bias is considered one of the most prevalent biases in LLM evaluation, RM training, and RLHF. This phenomenon arises from the confluence of multiple contributing factors in the output. For example, the preference for the model's use of listing, markdown style, URL, or complex tokens can all contribute to length bias. Thus, calibrating the length bias can simultaneously mitigate other more basic biases to some extent.
>
> Existing works focusing on mitigating length bias often require extra data [1], models [2], and training [3]. Or they are limited in their focus on the alignment algorithms [4]. Compared to them, our proposed method can be seamlessly applied to different scenarios and is very efficient regarding computation and data.
>
> [1] Offsetbias: Leveraging debiased data for tuning evaluators
>
> [2] Reward model ensembles help mitigate overoptimization. ICLR 2024
>
> [3] Safe RLHF: safe reinforcement learning from human feedback. ICLR 2024
>
> [4] Disentangling length from quality in direct preference optimization. ACL 2024

---

> ### Author Response · Authors · 2024-11-18
> **Responses to reviewer fLWP Part 3/3**
>
> >  Bias and Spurious Correlation: How does the characteristic bias lead to spurious correlations? Examples would facilitate a better understanding for the reader.
>
> The spurious correlation in our submission indicates that the reward model, trained on the human preference dataset, may overly rely on some shortcuts / superficial features (e.g., the length of the output) to determine the preference.
>
> > Hyperparameter Selection: What guidelines are there for selecting hyperparameters in different scenarios, such as $\beta_0$ and \$beta\1$?
>
> The only hyper-parameter for the proposed RC-LWR method is bandwidth $f$, which determines the portion of the dataset we used for regression given a point of interest. The bandwidth was set as the default as in the statemodel.api implementation for all reward models and tasks. We also study the sensitivity of the proposed method to the bandwidth choice (Fig. 7\) and verify that the calibration improvement is robust to the different bandwidths. We are sorry that algorithm 1 in the appendix may be misleading. $ \beta_1$ and $\beta_0$ are not hyper-parameters but are fitted slope and intercept of the LWR,
>
> > Fig. 5 x-axis: What does the x-axis represent in Figure 5? Providing clarity would help readers interpret the results more effectively.
>
> As mentioned in the caption of Fig. 5, the x-axis shows the original performance for different reward models on the reward bench, and the y-axis shows the calibrated performance. Thanks for your suggestion. We have revised the caption for clarity.
>
> >  There are several methods that can mitigate length bias, particularly in DPO. Have you considered simple baselines such as simPO?
>
> As discussed in the related work section in our submission, existing works that mitigate (length) bias are mainly from the perspective of (a) data handling, (b) reward model engineering, and (c) alignment algorithms. (a) and (b) usually require extra data collection and reward model training and is not post-hoc. And (c) is specifically designed for alignment procedure, thus having limited application scenarios. For example, using the length regularization term in SimPO for the RM-as-LLM-evaluator setting is not straightforward.
>
> Regarding comparison with SimPO, our method **achieves comparable results to simPO while using a weaker reward model.**
>
> Specifically, we use the same LLM (Meta-Llama-3-8B-Instruct and gemma2-9b-it), the same prompt dataset (Ultrafeedback), and the same sampled responses from the model. The only difference is that we use weaker reward models (FsfairX-LLaMA3-RM-v0.1 with 84.4 on rewardbench, GRM-llama3-8B-sftreg with 87.0 on reward bench)  to annotate the preference, and simPO employs a stronger one (ArmoRM-Llama3-8B-v0.1, 90.4 on RewardBench). Detailed results are shown in the Table below:
>
> | Method - LLM - RM - RM performance on RewardBench | AlpacaEval 2.0 Winrate |
> | :---- | :---- |
> | SimPO-Llama3-ArmoRM - 90.7                        | 53.7                   |
> | SimPO-gemma2-ArmoRM - 90.7                        | 72.4                   |
> | RC-LWR-Llama3-Fsfairx - 84.4                      | 51.4                   |
> | RC-LWR-Llama3-GRM - 87.0                          | 50.5                   |
> | RC-LWR-gemma2-Fsfairx - 84.4                      | 69.5                   |
> | RC-LWR-gemma2-GRM - 87.0 | 70.5 |
>
> We hope our responses could address your questions. Please let us know if you have further concerns or want to discuss more!

---

> ### Author Response · Authors · 2024-11-22
> **Request for discussion**
>
> Dear reviewer fLWP,
>
> We have posted some experiment results (e.g., a comparison with the RM engineering method) and clarifications to answer your questions. We have also updated our paper according to your suggestions. We would appreciate it if you could let us know whether our responses properly address your concerns.
>
> Look forward to your reply.
>
> Best regards,
>
> Authors

---

> > ### Comment · Reviewer_fLWP · 2024-11-23
> >
> > Thank you for your response. My concerns have been addressed. I have raised my score.

---

### Official Review · Reviewer_Lf2Z · 2024-11-04

**Soundness:** 3
**Presentation:** 2
**Contribution:** 2
**Rating:** 8
**Confidence:** 4

**Summary:**

This paper focuses on correction of spurious correlations in LLM-based reward models learned from preference data. The authors assume that the spurious feature (e.g., length) biases the rewards in a consistent away. They propose to estimate this consistent bias via a locally-weighted-regression method. Then, the estimated bias can be subtracted from learned reward in order to achieve a model without the spurious correlation. The authors focus on length as the spurious feature and compare to a baseline which subtracts a constant factor times the length from the reward. They evaluate a variety of reward models with various adjustments to eliminate length bias across several metrics, including RewardBench scores, correlation with human rankings of LLMs in ChatbotArena, and length-controlled win rate in AlpacaEval2. In general, the proposed method appears to better remove length bias.

**Strengths:**

The proposed method is relatively simple and is well-motivated conceptually. The fact that it can be applied post-hoc means that retraining is not required, which is helpful. I am not very familiar with the literature in this area but to my knowledge it is also novel. The results are extensive and seem to generally support that post-hoc reward calibration effectively removes length bias.

**Weaknesses:**

The main weaknesses of the paper include:

 * **Presentation:** I found the introduction of the method in Section 3 to be relatively clear, but the experiments in Sections 4 and 5 were much more difficult to understand. While I eventually could figure out the gist of the experiment design, it is often quite complicated and could be better presented. In particular, Section 4.2 is very dense. It describes multiple evaluation methods (AlpacaEval1, AlpacaEval2, ChatbotArena), which if I understand correctly in some cases used as both gold standard (AlpacaEval2) and as baselines (LC). The combination of many reward models, LLMs, calibration methods, and comparison rankings makes it quite important to carefully describe the purpose behind the experimental setup and why it is a good evaluation of the calibration methods. I think the current text requires several readings to understand these details. The rest of Sections 4 and 5 are also very dense and difficult to understand at first. It would be good to improve the presentation of these such that the results are easier to understand.

 * **Lack of insight into the result of reward calibration:** while the authors present many experiments on how reward calibration affects downstream performance, they do not present what the actual calibration process looks like for typical reward models. It would be helpful to see what the actual $\hat{r}\_\theta(c)$ function looks like over a range of values for $c$ for various reward models. This could help readers understand why RC-Mean and RC-LWR outperform length penalty, especially if the $\hat{r}\_\theta(c)$ function is quite non-linear. On the other hand, if it is close to linear, then it would raise the question of why not simply tune the constant $\alpha$ in the linear length penalty. Perhaps the length penalty simply does not use the right coefficient—in fact, it seems that it should always be possible to remove the length correlation measured in Table 3 by using a linear length penalty with the right coefficient.

 * Finally, the authors do not consider the extent to which the correlation between length and reward may not be entirely spurious—there may actually be some amount of weight that the reward models *should* place on length.

Small issues:
 * "LLMs alignment" -> "LLMs' alignment"
 * Line 470 "board" -> "broad"

**Questions:**

With regards to the second weakness above, it would be helpful if the authors could plot the learned $\hat{r}_\theta(c)$ function for several reward models to see what kind of affect subtracting it has compared to the linear length bias $\alpha \\, c$. Also, it would be helpful to compare to a baseline of a linear length bias where the factor $\alpha$ is tuned per reward model.

---

> ### Author Response · Authors · 2024-11-18
> **Response to reviewer Lf2Z**
>
> Thanks for your invaluable suggestions. We hope that our revised paper and our pointwise responses address your concerns.
>
> >Presentation
>
> Thanks for your suggestions. The presentation is dense because we tried to evaluate our method comprehensively in at least three scenarios, which requires compressing a lot of information into our paper. To make it more readable, we have paraphrased and revised Section 4.2 of our paper and will further clarify the presentation of our experiment sections.
>
> > Lack of insight into the result of reward calibration
>
> Thanks for your valuable suggestion. To provide more insights into the results of our calibration, we supplement figures about reward w.r.t. length before and after calibration in the appendix of our revised paper.
>
> Overall, the Spearman correlation decreases to nearly 0 after RC-LWR calibration. Moreover, the correlation between reward and length is non-linear, making the locally weighted regression a more robust and general form for estimating and removing the bias of interest. After length-penalty calibration, the overall correlation of the entire dataset will decrease, but locally, the bias may be amplified.
>
> > The authors do not consider the extent to which the correlation between length and reward may not be entirely spurious—there may actually be some amount of weight that the reward models should place on length.
>
> Please refer to the common concern in the General Comment. We found that introducing a hyper-parameter $\gamma$ to determine how much of the estimated bias term to subtract from the original bias reward could smoothly control the calibration effect. What is more, one can tune this $\alpha$ to achieve better rewarding performance if some prior knowledge about the dataset is known/estimated.
>
> > With regards to the second weakness above, it would be helpful if the authors could plot the learned $\hat{r}\_\theta(c)$ function for several reward models to see what kind of effect subtracting it has compared to the linear length bias $\alpha$ c.
>
> Thanks for your suggestion! We have supplemented the relevant figures (Fig. 17 -- 26) in the appendix of our revised paper. Please see the descriptions above and in the revised paper.
>
> >  Also, it would be helpful to compare to a baseline of a linear length bias where the factor $\alpha$ is tuned per reward model.
>
> Thanks for the suggestion. We supplement an ablation comparison in a setting where hyper-parameters could be tuned for every reward model. Specifically, we select the optimal $\alpha$ for every RM in length-penalty baseline and optimal $\gamma$ for every RM of our method. In this setting, the length-penalty baseline achieves **3.27** performance gain, and RC-LWR achieves **3.67**.  Please see our response to reviewer PP1Zs' weakness 2 for more ablations about $\alpha$ values.
>
> Robustness to hyper-parameters is a highly desirable property in practice. We want to highlight that length-penalty is sensitive to $\alpha$, and our method is more robust to hyper-parameter choices (Fig. 7 and Fig. 8). And because $\alpha$ is not only reward-model-dependent but is also task-dependent, it is non-trivial to select optimal $\alpha$ per RM and task.
>
> Please let us know what else we can do to help you to recommend our paper. Thanks for your time.

---

> > ### Comment · Reviewer_Lf2Z · 2024-11-21
> > **Response to rebuttal**
> >
> > Thank you for the thorough response to my review. I have raised my score and advocate for acceptance. I would still encourage the authors to think through ways of making Section 4 easier to read, although I appreciate that it is difficult to convey so much detail in the page limit.

---

> > > ### Author Response · Authors · 2024-11-21
> > > **Response from authors**
> > >
> > > Thanks for raising your score! We will continually optimize the presentation of Section 4 for clarity!

---

### Official Review · Reviewer_PP1Z · 2024-11-04

**Soundness:** 2
**Presentation:** 3
**Contribution:** 2
**Rating:** 6
**Confidence:** 4

**Summary:**

This paper proposes a novel method, “Post-hoc Reward Calibration,” to mitigate biases in reward models (RMs), particularly focusing on LLM output text length bias in RMs for large language models (LLMs). The method removes this bias by calibrating the reward score without additional RM retraining, making it computationally efficient. The approach assumes independence between the true reward and text length, constructs a locally linear regression model from text length to (proxy) reward values using the Robust Locally Estimated Scatterplot Smoothing (LOWESS), and calibrates the (proxy) reward with this regression model. The authors evaluate the approach on three tasks—RewardBench, RM as LLMs Evaluator, and RM for LLMs Alignment—and demonstrate that the calibrated RMs better align with human evaluations by providing fairer, length-independent scoring.

**Strengths:**

**Originality**: The paper proposes a unique, post-hoc calibration approach to mitigate reward model biases related to LLM output length without requiring additional data or retraining.

**Efficiency**: This method is computationally efficient, applying bias correction as a post-processing step, making it easily applicable across tasks without modifying reward models.

**Empirical Validation**: The approach is thoroughly evaluated on three tasks, effectively demonstrating its capacity to mitigate length bias.

**Weaknesses:**

**Assumption**:
The assumption of “Independence of the biased characteristic” may be overly restrictive in practical applications, especially given that this study focuses on text length as the biased characteristic. My understanding of this assumption (line 166) is that even when $x_1$ and $x_2$ are fixed at any arbitrary lengths $c_1$ and $c_2$, respectively, the true reward difference between $x_1$ and $x_2$ would be expected to be zero, implying no correlation between reward and text length. However, in practical applications, this assumption will often not hold. For example, in cases where prompts explicitly request brevity or detail, such as “respond concisely” or “provide a detailed explanation,” reward often depends on the response length. This assumption could thus limit the generalizability of the proposed method, especially for tasks where length naturally influences quality. A more flexible approach that adjusts for prompt-specific length expectations could enhance applicability.

**Baseline setting**:
The Length Penalty hyperparameter $\alpha$ seems to be fixed at 0.001, which may not be optimal across different reward models (RMs) and tasks due to variations in scale. While the proposed method benefits from adaptive $d$ tuning, the baseline might perform better with an α adjusted per RM or task, ensuring a fairer comparison. For instance, $\alpha$ could be proportional to the RM’s reward scale.

**Questions:**

How is the sensitivity of the length penalty hyperparameter to the performance compared to the proposed methods?

---

> ### Author Response · Authors · 2024-11-18
> **Responses to reviewer PP1Z Part 1/2**
>
> Thanks for your valuable comments and questions\! Please see our responses below point by point.
>
> > 1. Assumption: The assumption of “Independence of the biased characteristic” may be overly restrictive in practical applications, especially given that this study focuses on text length as the biased characteristic. My understanding of this assumption (line 166\) is that even when x1 and x2 are fixed at any arbitrary lengths c1 and c2, respectively, the true reward difference between x1 and x2 would be expected to be zero, implying no correlation between reward and text length. However, in practical applications, this assumption will often not hold. For example, in cases where prompts explicitly request brevity or detail, such as “respond concisely” or “provide a detailed explanation,” reward often depends on the response length. This assumption could thus limit the generalizability of the proposed method, especially for tasks where length naturally influences quality. A more flexible approach that adjusts for prompt-specific length expectations could enhance applicability.
>
> Thanks for your question; please see our general comments for the common concern. Basically, we highlight that the proposed method achieves good performance gains across various reward models and settings, where the assumption may not perfectly hold (e.g., for RewardBench, the ideal correlation between reward and length should be slightly negative). Then, we emphasise that a hyper-parameter $\gamma$ can be introduced to control the calibration effect, which helps users preserve or even inject certain biases into the reward model.
>
> We also consider investigating prompt-dependent calibration an exciting research direction (though modelling the correlation between prompt and bias characteristics may require more data and computation, thus introducing a trade-off between efficiency and effectiveness). We would like to explore it in our future work.

---

> > ### Author Response · Authors · 2024-11-18
> > **Responses to reviewer PP1Z Part 2/2**
> >
> > > 2. Baseline setting: The Length Penalty hyperparameter is fixed at 0.001, which may not be optimal across different reward models (RMs) and tasks due to variations in scale. While the proposed method benefits from adaptive \alpha tuning, the baseline might perform better with an \alpha adjusted per RM or task, ensuring a fairer comparison. For instance, it could be proportional to the RM’s reward scale.
> >
> > Thanks for your comments! Please see our pointwise responses to your concern:
> >
> > Regarding Ensuring a fairer comparison, we respectfully believe that the comparison here is fair because we keep the hyper-parameter consistent for all 33 RMs in the Rewardbench setting and for all 10 RMs in the RM-as-LLM-evaluator setting for our proposed method. The baselines. hyperparameter (0.001) was adopted from prior work. We validated this value via ablation analysis on RewardBench as below.
> >
> > To further consolidate our experimenting results, we provide more fine-grained hyper-parameter analyses for the length-penalty method in the RewardBench benchmark setting.
> >
> > 1. First, we show that choosing $\alpha=0.001$ following previous works gives better performance than other $\alpha$ values searched, including [0.1 0.05 0.02 0.01 0.005 0.002 0.001 0.0005 0.0002 0.0001]
> >
> >  | 0.0001 | 0.0002 | 0.0005 | 0.001    | 0.002 | 0.005 | 0.01  | 0.02  | 0.05   | 0.1    |
> >  | ------ | ------ | ------ | -------- | ----- | :---: | ----- | ----- | ------ | ------ |
> >  | 0.37   | 0.73   | 1.52   | **2.18** | 1.92  | -2.07 | -6.14 | -9.58 | -12.36 | -13.60 |
> >
> > 2. Then, regarding the suggestion that "$\alpha$ could be proportional to RMs' reward scale", we normalise the reward with the Z-score normalisation to decouple the effect of different reward scales and then try different $\alpha$ values across all models, we found that the optimal $\alpha$ change, and the overall performance slightly improves but still underperforms RC-LWR (3.14 according to Fig.  8).
> >
> >  | 0.0001 | 0.0002 | 0.0005   | 0.001 | 0.002  | 0.005 | 0.01   | 0.02   | 0.05   | 0.1    |
> >  | ------ | ------ | -------- | ----- | ------ | :---: | ------ | ------ | ------ | ------ |
> >  | 0.93   | 1.56   | **2.75** | 1.67  | -2.673 | -8.03 | -11.27 | -13.29 | -14.87 | -15.68 |
> >
> > 3. Oracle hyper-parameter selection setting: if we select the optimal $\alpha$ values for each model and then average it across all 33 RM models, we can get **3.27** performance gain. Similarly, for our proposed method we try different $\gamma$ values from [0.6 0.7 0.8 0.9 1.0 1.1 1.2 1.3 1.4 1.5] (for $\gamma$ please refer to common concern 1 or Eq. 8 in our submission), we can get **3.67** performance gain.
> >
> > Importantly, we believe **the robustness to hyper-parameters is essential in practical use**. It's non-trivial to adjust $\alpha$ per RM and task. On the contrary, one can efficiently utilise our method regardless of the reward distribution of the RM and the length distribution of the task (which may vary a lot; please refer to Fig. 9 for details). Moreover, we validated that our method is robust to hyper-parameter bandwidth in Fig. 7, showcasing that our method demonstrates stronger practicability compared with the length penalty baseline.
> >
> > To conclude, we agree with the reviewer that the length penalty could yield better results if one could search optimal $\alpha$ per RM and task. However, we respectfully believe that the current comparison is still fair. We supplement analysis on RewardBench (line 1180 - 1120) to demonstrate that the performance of length penalty baseline is sensitive to $\alpha$, while our method robustly adapts to different RMs and tasks.
> >
> > > How is the sensitivity of the length penalty hyperparameter to the performance compared to the proposed methods?
> >
> > According to the results above, the penalty method is sensitive to this hyper-parameter.
> >
> > Thanks for your comments. Please let us know if you have any other concerns or want further discussion.

---

> ### Author Response · Authors · 2024-11-22
> **Request for feedback**
>
> Dear reviewer PP1Z,
>
> We have posted some experiment results to answer your questions about **Assumption and Baseline Settings**. We have also updated our paper to include extra results. We would appreciate it if you could let us know whether our responses address your concerns.
>
> Look forward to your reply.
>
> Best regards,
>
> Authors

---

> > ### Comment · Reviewer_PP1Z · 2024-11-22
> >
> > Thank you for your detailed and thoughtful response. The clarifications on the assumption of independence and the justification for the hyperparameter settings are convincing. Based on this, I have raised my score accordingly.

---

### Author Response · Authors · 2024-11-18
**General Comment: Common concern and major revisions**

We sincerely appreciate all the reviewers' time and effort in reviewing our paper. We are glad to find that reviewers generally recognise our strengths/contributions:

- We proposed a reward calibration method to correct bias in already-trained reward models, the proposed method is novel (**PP1Z**, **Lf2Z**) and is demonstrated to be effective for removing length bias (**PP1Z, Lf2Z, fLWP, aRiJ**).
- Our proposed method is computationally efficient (30 seconds for calibrating 300k samples with one CPU, **PP1Z, Lf2Z, fLWP, aRiJ**), and it does not require any additional data (**PP1Z**) or reward model retraining (**PP1Z, Lf2Z**)
- We evaluate our proposed method in three settings – RewardBench, RM as LLMs, and RM for LLMs alignment (**PP1Z, Lf2Z, aRiJ**), providing extensive/thorough empirical results to confirm the effectiveness of our proposed method(**PP1Z, Lf2Z**)

We thank the invaluable suggestions from reviewers, which helped a lot in the further improvement of this paper. In addition to the pointwise responses below, we try to address the reviewer's common concern as follows:

> **Common Concern: Reviewers (PP1Z, fLWP, Lf2Z) are concerned that the assumption of independence between the underlying reward and the bias characteristics may sometimes not hold, which may hurt the proposed method’s generality.**

We first want to highlight that our method still achieves significant improvements under various scenarios where the independence assumption **may not** perfectly hold: (a) **33** RMs on RewardBench; (2) **10 RMs + GPT4-as-judge** for the LLM-evaluation setting; and (3) various LLM--RM combinations for RLHF. Our experimental results firmly support the generality of our method.

We agree with the reviewer that the independence assumption may sometimes be violated. So, we highlight in line 163 that we consider a “general” prompt dataset for this assumption. Furthermore, in the appendix of our submission, we introduce a simple yet effective method to control the calibration effect: using a hyper-parameter $\gamma$ to determine how much of the estimated bias will be removed from the original reward margin. The form with $\gamma$ is (Eq. 8 in our submission):

$\hat{\Delta}^*_{r_\theta}=\Delta_{r_\theta}(x_1,x_2)-\gamma\times (\hat{r}_\theta(|x_1|)-\hat{r}(|x_2|))$

Results in the appendix (Fg. 8 and Tab. 4\) show that $\gamma$ could smoothly control the calibration effect (help you preserve or inject some bias). A noteworthy statistic for the RewardBench is that selecting the shorter response yields a 60% accuracy. Thus, the ideal correlation between rewards and response length should be slightly negative. The results in Fig. 8 support the intuition: if $\gamma=1$ results in zero correlation after calibration, the optimal $\gamma$ for RewardBench should exceed 1\. Our findings suggest that the best average performance gain occurs when $\gamma= 1.4$.  The introduced $\gamma$ enhances the practicability of the proposed method, as one can adjust $\gamma$ to achieve better rewarding performance if specific prior knowledge about the correlation between rewards and the characteristic of interest is known.

Overall, we argue that our method is practical and supports a wide range of application scenarios because:

1. our method **does not** introduce any computational overhead and can be easily implemented
2. Most reward models are overly biased by lengthy outputs, and calibrating them for a general domain mostly brings improvements;
3. A constant hyper-parameter $\gamma$ can be introduced to control the calibration effect smoothly.

In response to this common concern, we highlight the
at we con introduce $\gamma$ introduced in our main paper (line 517 and line 537).

We also revised our paper following suggestions from reviewers. All revised parts are marked as blue.

**Major revisions include**:

1. **Improved presentation**: Following the suggestion of reviewer Lf2Z,  we revised Section 4, especially Section 4.2, to better convey the essence of our experiment design.
2. **Extra Analysis**: Reviewers PP1Z and LF2Z asked about the Length-Penalty baseline (e.g., the sensitivity to hyper-parameters). We thus have provided an analysis of the length-penalty baseline, demonstrating its sensitivity to hyper-parameters.
3. **Additional baseline**: Following reviewer flWP's suggestion, we compare our calibration method with the model ensemble approach. Our results demonstrate that RC-LWR outperforms ensemble methods except when ensembled with a much stronger model.

1. **Demonstrations**: Following the suggestions from reviewer Lf2Z, we have supplemented the appendix of our revised paper with figures (Fig. 17—Fig. 26) about reward with length before and after calibration to provide insight into the calibration results.

We once again thank reviewers for their insightful reviews, and don't hesitate to contact us if we can do something else to help you better understand and recommend our paper.

---

### Meta-Review · Area_Chair_R2PW · 2024-12-22

**Metareview:**

This paper studies the length bias in reward modeling and RLHF, which is a well-known issue in learned reward models and language model preferences. They provide a method of "Post-hoc Reward Calibration" to address it and obtained promising empirical results.

Strengths:
The studied problem is well-motivated. The proposed approach is simple and needs no extra training. Experiments also support the proposed argument.

Weaknesses:
The proposed approach assumes independence between the true reward and text length, which appears too strong. The compared baselines are also not sufficiently challenging.

Most of the weaknesses have been addressed during the rebuttal period. All reviewers suggest acceptance for this paper, and I agree with their assessment.

**Additional Comments On Reviewer Discussion:**

During the rebuttal period, the authors provided additional experiments and explanations, and addressed most of the concerns from the reviewers.

---

### Decision · Program_Chairs · 2025-01-22

Accept (Poster)